# Construction of Pd-Zn dual sites to enhance the performance for ethanol electro-oxidation reaction

Yajun Qiu[1], Jian Zhang [2✉], Jing Jin[3], Jiaqiang Sun[4], Haolin Tang[5], Qingqing Chen[6], Zedong Zhang[1], Wenming Sun [7✉], Ge Meng[2], Qi Xu[1], Youqi Zhu[8], Aijuan Han[3], Lin Gu [9], Dingsheng Wang [1✉] & Yadong Li [1]

Rational design and synthesis of superior electrocatalysts for ethanol oxidation is crucial to practical applications of direct ethanol fuel cells. Here, we report that the construction of Pd-Zn dual sites with well exposure and uniformity can significantly improve the efficiency of ethanol electro-oxidation. Through synthetic method control, Pd-Zn dual sites on inter-metallic PdZn nanoparticles, Pd-Pd sites on Pd nanoparticles and individual Pd sites are respectively obtained on the same N-doped carbon coated ZnO support. Compared with Pd-Pd sites and individual Pd sites, Pd-Zn dual sites display much higher activity for ethanol electro-oxidation, exceeding that of commercial Pd/C by a factor of ~24. Further computational studies disclose that Pd-Zn dual sites promote the adsorption of ethanol and hydroxide ion to optimize the electro-oxidation pathway with dramatically reduced energy barriers, leading to the superior activity. This work provides valuable clues for developing high-performance ethanol electro-oxidation catalysts for fuel cells.

[1] Department of Chemistry, Tsinghua University, Beijing, China. [2] College of Chemistry and Materials Engineering, Wenzhou University, Wenzhou, Zhejiang, China. [3] State Key Laboratory of Chemical Resource Engineering, Beijing University of Chemical Technology, Beijing, China. [4] State Key Laboratory of Coal Conversion, Institute of Coal Chemistry, Chinese Academy of Sciences, Taiyuan, Shanxi, China. [5] State Key Laboratory of Advanced Technology for Materials Synthesis and Processing, Wuhan University of Technology, Wuhan, China. [6] Key Laboratory of Functional Molecular Solids, Ministry of Education, College of Chemistry and Materials Science, Anhui Normal University, Wuhu, Anhui, China. [7] College of Science, China Agricultural University, Beijing, China. [8] Research Center of Materials Science, Beijing Institute of Technology, Beijing, China. [9] Institute of Physics, Chinese Academy of Sciences, Beijing, China. ✉email: zhangjian1209@outlook.com; swm@cau.edu.cn; wangdingsheng@mail.tsinghua.edu.cn

Direct ethanol fuel cells (DEFCs) have attractive application prospect in the fields of power generation, fixation, and transformation for their good agreement with the urgent demand of current energy and environment[1–15]. The development of highly active and low-cost electrocatalysts for the ethanol oxidation reaction (EOR) lays at the heart of DEFCs[16–25]. Downsizing of metal nanoparticles into individual metal sites is commonly regarded as an efficient approach to enhance the performance and cost-effectiveness of electrocatalysts, but may be improper for specific reactions catalyzed by multiple sites or peculiar active centers[26–31]. For now, alloying of Pd with base metals (M) like Zn, Ni, Sn, Co, Cu, Ge, etc. to construct Pd-M dual sites, which are more active for EOR than pure Pd sites[32–43]. The superior activity of such dual sites is generally considered to arise from base metals tuning the electronic property of Pd or synergistically taking part in the reaction[44,45]. The ambiguous role of the dual sites is the major obstacle to creating promising electrocatalysts for EOR[46–48]. In addition, in order to increase the exposure and uniformity of dual sites for better performance, common solvothermal methods are utilized to synthesize small-sized intermetallic compounds but usually cause residual surfactants covering dual sites with insufficient EOR performance[49–52]. Hence, it is highly desirable to develop efficient means to establish exposed and uniform Pd-M dual sites for prominent EOR electrocatalysts, and clarify their true principle of promotion effect for the rational design of new catalysts.

Herein, we demonstrate an effective strategy to fabricate Pd–Zn dual sites for efficient EOR with Pd–Pd sites and individual Pd sites as references for investigating the mechanism of promotion effect. Volatile Zn atoms from ZnO nanorods can bound Pd atoms on N-doped carbon to form well exposed and uniform Pd–Zn dual sites. The Pd–Zn dual sites make the catalyst (PdZn/NC@ZnO) perform the much higher mass activity than the other two catalysts with Pd–Pd sites ($Pd_n$/NC@ZnO) and individual Pd sites ($Pd_1$/NC@ZnO), and even commercial Pd/C by ~24 times. Further DFT calculations manifest that Pd–Zn dual sites are beneficial to adsorbing ethanol and hydroxide ion with lower reaction energy, tuning the reaction pathway of EOR with significant reduced energy barriers in contrast to Pd–Pd and individual Pd sites, which accounts for the improved activity of PdZn/NC@ZnO.

## Results

**Synthesis and characterization of the catalysts.** The fabrication of PdZn/NC@ZnO is realized by a polydopamine (PDA) confined and Zn vapor-assisted strategy (Fig. 1a). First, ZnO nanorods are pre-synthesized as the support via the reported method (Fig. 1b)[53]. Then, the support is coated with a regular layer of PDA (~10–20 nm) as is observed by transmission electron microscopy (TEM) (Fig. 1c and Supplementary Fig. 1). Next, $Pd(OH)_2$ is loaded on the PDA layer through slowly precipitating Pd precursors with alkali, which is confirmed by X-ray diffraction (XRD) and TEM detections showing no obvious nanoparticles (NPs) (Supplementary Fig. 2). Under high-temperature reduction with $H_2$, the PDA layer transforms into N-doped carbon (NC) to restrict the migration of Pd atoms, and the Zn vapor is generated from ZnO for bounding Pd atoms to form Pd–Zn dual sites, finally leading to intermetallic PdZn NPs as evidenced by the scanning transmission electron microscopy (STEM) (Fig. 1d). The statistical distribution reveals the average particle size of such PdZn NPs being as small as 4.75 ± 0.77 nm (the inset in Fig. 1d and Supplementary Fig. 3). The size of PdZn NPs being small can be also reflected by the XRD detection which shows no PdZn signals except for neat ZnO peaks (Supplementary Fig. 4). After the ZnO being etched away, there are still PdZn NPs on the NC

carrier in Supplementary Fig. 5a, and only graphitic carbon peaks are observed in the XRD patterns because of the PdZn NPs being too small (Supplementary Fig. 5c). When the loading of Pd precursor is increased five times, it can be seen that PdZn NPs become larger and exhibit featured intermetallic PdZn signals in the XRD patterns (Supplementary Fig. 5b, c), which suggests the nature of the PdZn NPs in PdZn/NC@ZnO being PdZn intermetallic compounds[54,55].

In order to confirm the composition of PdZn NPs, energy-dispersive X-ray spectroscopy (EDS) line scanning measurement is first carried out (Fig. 1e). The profile displays that X-ray counts of Pd and Zn fluctuate in a similar manner, indicating the simultaneous presence of Pd and Zn atoms in the nanoparticle. Further atomic-resolution EDS elemental mapping analysis manifests that Pd and Zn homogeneously dispersed over the entire nanoparticle (Fig. 1f and Supplementary Fig. 6). The typical aberration-corrected high-angle annular dark-field scanning transmission electron microscope (AC HAADF-STEM) image of one nanoparticle exhibits clear bright dots, within which the heavier Pd atoms (bright) can be distinguished from Zn atoms (darker) (Fig. 1g). More importantly, the ordering of these bright dots accords with the atomic arrangement of (002) plane in the intermetallic PdZn (P4/mmm) crystal structure, and the lattice spacing of 2.1 and 2.9 Å also agree with inter-plane distances of (200) and (110) planes (Fig. 1g and Supplementary Fig. 7). Besides, the intermetallic PdZn (111) planes can be also observed on the nanoparticle by high-resolution transmission electron microscopy (HR-TEM) (Supplementary Fig. 8). These results testify the intermetallic nature of PdZn NPs with uniform Pd–Zn dual sites in PdZn/NC@ZnO. In contrast, when treated at low temperature (400 °C) to avoid the Zn vapor, only smaller Pd NPs (2.75 ± 0.66 nm) with Pd–Pd sites are formed in the catalyst ($Pd_n$/NC@ZnO) as deduced from STEM, TEM, and EDS line scanning tests (Fig. 1h, Supplementary Figs. 9–12). Moreover, individual Pd sites can be even obtained on the same support ($Pd_1$/NC@ZnO) through the milder synthetic condition (200 °C), as determined by the solely bright dots assigned to isolated Pd atoms showed in the AC HAADF-STEM image (Fig. 1i and Supplementary Figs. 13–15). The smaller Pd NPs and individual Pd sites testify the PDA layer confining the aggregation of Pd atoms. The loading amount of Pd is 0.30, 0.40, and 0.35 wt% for PdZn/NC@ZnO, $Pd_n$/NC@ZnO, and $Pd_1$/NC@ZnO as affirmed by inductively coupled plasma optical emission spectrometry (ICP-OES) measurements.

The different dispersion of Pd in three samples can be also verified by X-ray absorption spectrometry techniques (XAS) (Fig. 2a). Fourier transformed extended X-ray absorption fine structure (FT-EXAFS) spectra at Pd K-edge show that the first nearest-coordination peak of Pd in PdZn/NC@ZnO locates at the smaller R value of 2.2 Å in comparison with that in Pd foil at 2.5 Å from the Pd–Pd contribution (for corresponding EXAFS in k-space, see Supplementary Fig. 16). The fitting analysis elucidates that this first shell peak is contributed from Pd–Zn (~2.6 Å) and longer Pd–Pd (~2.8 Å) bonds, coinciding with the atomic structure of intermetallic PdZn compound (Supplementary Fig. 17 and Supplementary Table 1). While for $Pd_n$/NC@ZnO and $Pd_1$/NC@ZnO, there is the only one peak from shorter Pd–Pd bond (2.5 Å) and Pd–N(C) bond (1.7 Å) in their FT-EXAFS curves (Supplementary Fig. 17 and Supplementary Table 1). These results further demonstrate Pd species existing as Pd–Zn dual sites, Pd–Pd, and individual Pd sites in three catalysts.

The electronic structure of three catalysts is next investigated through Pd K-edge X-ray absorption near edge structure (XANES) spectroscopy. The adsorption edge energy ($E_0$) of PdZn/NC@ZnO is higher than that of Pd foil and $Pd_n$/NC@ZnO

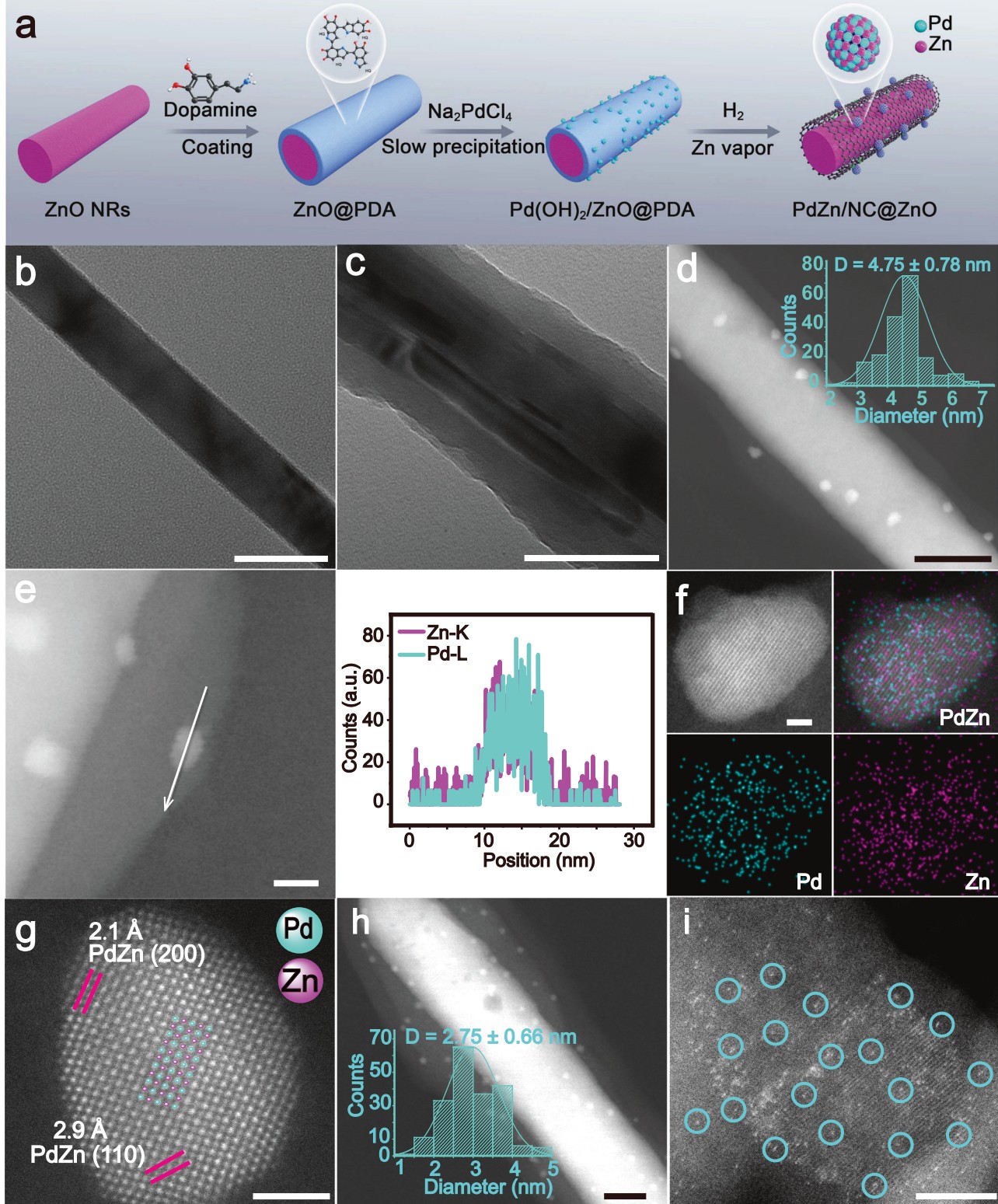

**Fig. 1 Synthetic scheme and characterizations of synthesized catalysts. a** Synthetic strategy of PdZn/NC@ZnO. **b** TEM image of ZnO nanorods. Scale bar, 50 nm. **c** TEM image of ZnO@PDA. Scale bar, 100 nm. **d** STEM image of PdZn/NC@ZnO, the inset is the size distribution histogram of PdZn nanoparticles in PdZn/NC@ZnO. Scale bar, 50 nm. **e** EDS line scanning profile across one PdZn nanoparticle in PdZn/NC@ZnO. The signals were collected from the Pd L edge and the Zn K-edge. Scale bar, 10 nm. **f** AC HAADF-STEM image of one PdZn nanoparticle in PdZn/NC@ZnO. Scale bar, 2 nm. **g** AC HAADF-STEM elemental mappings of one PdZn nanoparticle in PdZn/NC@ZnO. Scale bar, 2 nm. **h** STEM image of Pd$_n$/NC@ZnO, the inset is the size distribution histogram of Pd nanoparticles in Pd$_n$/NC@ZnO. Scale bar, 20 nm. **i** AC HAADF-STEM image of Pd$_1$/NC@ZnO. Scale bar, 5 nm.

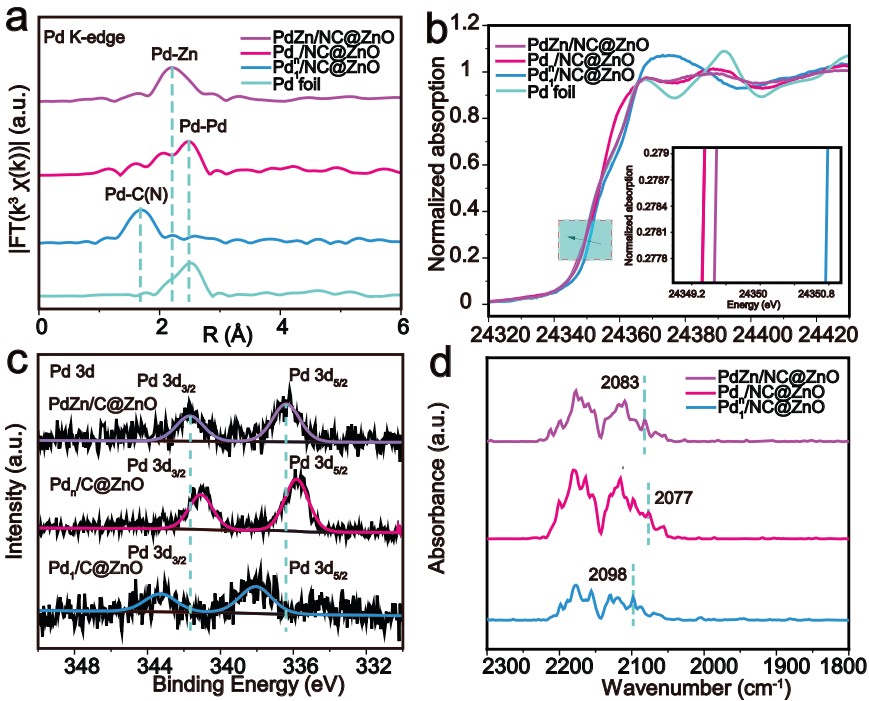

**Fig. 2 X-ray absorption spectroscopy characterization of the catalysts. a** FT-EXAFS spectra at the Pd k-edge of PdZn/NC@ZnO, $Pd_n$/NC@ZnO, and $Pd_1$/NC@ZnO with Pd foil as the reference. **b** XANES spectra at the Pd k-edge of PdZn/NC@ZnO, $Pd_n$/NC@ZnO, and $Pd_1$/NC@ZnO with Pd foil as the reference. **c** XPS spectra in Pd 3d region of PdZn/NC@ZnO, $Pd_n$/NC@ZnO, and $Pd_1$/NC@ZnO. **d** DRIFT spectra for PdZn/NC@ZnO, $Pd_n$/NC@ZnO, and $Pd_1$/NC@ZnO.

but much lower than that of $Pd_1$/NC@ZnO (Fig. 2b), indicating that the electron density of Pd decreases sightly for PdZn/NC@ZnO but significantly for $Pd_1$/NC@ZnO with respect to $Pd_n$/NC@ZnO. This conclusion can be also drawn from X-ray photoelectron spectroscopy (XPS) and CO diffuse reflection infrared Fourier transform spectroscopy (CO-DRIFTS) studies. It can be evidenced from XPS spectra that Pd 3d signal of PdZn/NC@ZnO locates rightly between that of $Pd_n$/NC@ZnO and $Pd_1$/NC@ZnO, revealing the moderate electron density of Pd in PdZn/NC@ZnO (Fig. 2c). Moreover, CO-DRIFTS experiments disclose that CO stretching frequency has a blue shift from 2077 cm$^{-1}$ for $Pd_n$/NC@ZnO to 2083 and 2098 cm$^{-1}$ for PdZn/NC@ZnO and $Pd_1$/NC@ZnO (Fig. 2d). This can be attributed to the linear combination of CO on Pd species with distinct electron deficiency in Pd–Zn dual sites and individual Pd sites in comparison with Pd–Pd sites. The varying decrease in electron density of Pd discloses the slight electron transfer from Pd to Zn atoms in Pd–Zn dual sites and the strong electron transfer from Pd to the carrier in individual Pd sites[56–59].

**Performance evaluation for EOR.** The performance for EOR is then evaluated on three catalysts with commercial Pd/C as the reference. In 1.0 M KOH and 1.0 M ethanol solution, cyclic voltammograms (CVs) reveal that PdZn/NC@ZnO exhibits the much higher current density than $Pd_1$/NC@ZnO, $Pd_n$/NC@ZnO, and Pd/C (Fig. 3a and Supplementary Fig. 18). Unlike the negligible mass activity of $Pd_1$/NC@ZnO and $Pd_n$/NC@ZnO, the mass activity of PdZn/NC@ZnO comes up to a high level as 18.14 A mg$_{Pd}^{-1}$, which is 24-fold to that of Pd/C (0.76 A mg$_{Pd}^{-1}$) and exceeds the performance of most Pd-based catalysts in the literature (Fig. 3b and Supplementary Table 2). It is noteworthy that the catalytic efficiency for EOR can even reach up to 77.51 A mg$_{Pd}^{-1}$ based on the surfaced Pd atoms of PdZn NPs in PdZn/NC@ZnO (Supplementary Fig. 19). The specific activity of

PdZn/NC@ZnO is 54.60 mA cm$^{-2}$, also surpassing that of Pd/C (1.48 mA cm$^{-2}$) by ~37 times (Fig. 3b). Electrochemical surface area (ECSA) for PdZn/NC@ZnO is determined to be 33.23 m$^2$ g$_{Pd}^{-1}$, smaller than that for $Pd_n$/NC@ZnO (36.76 m$^2$ g$_{Pd}^{-1}$), $Pd_1$/NC@ZnO (76.77 m$^2$ g$_{Pd}^{-1}$), and Pd/C (51.02 m$^2$ g$_{Pd}^{-1}$) (Fig. 3c and Supplementary Fig. 20)[60]. This implies the great intrinsic activity of active Pd–Zn sites in PdZn/NC@ZnO for EOR. Besides, the catalytic stability of PdZn/NC@ZnO is probed via chronoamperometry (CA) tests. It presents that PdZn/NC@ZnO remains larger current density all the time during the test than Pd/C which decays rapidly (Fig. 3d), representing a superior long-term activity for EOR. Moreover, CV curves of potential cycling displays negligible loss in peak current density of PdZn/NC@ZnO before and after 2000 potential cycles (Supplementary Fig. 21). Barely no residual Pd and Zn are detected in the reaction mixture, and the content of Pd in PdZn/NC@ZnO keeps the same as 0.3 wt % after the stability test (Supplementary Table 3). Corresponding characterizations of PdZn/NC@ZnO after potential cycles show no observable changes in the morphology and composition (Supplementary Figs. 22–24). These results prove the desired stability of PdZn/NC@ZnO for EOR. Obviously, the delightful activity and durability of PdZn/NC@ZnO clarifies that Pd–Zn dual sites are significantly superior to Pd–Pd and individual Pd sites for EOR.

The performance of PdZn/NC@ZnO for EOR is further evaluated in an actual alkaline membrane fuel cell. The polarization and power density curves in single DEFC with different electrocatalysts as anodes are shown in Supplementary Fig. 25. The open-circuit voltage of the fuel cell containing PdZn/NC@ZnO and Pd/C electrocatalysts is 0.846 and 0.843 V, respectively. The DEFC performances of PdZn/NC@ZnO (72.30 mW cm$^{-2}$) are better than that of Pd/C (64.75 mW cm$^{-2}$). Notably, when normalizing the power density values of the DEFC with respect to the Pd loading, the electrical performance of PdZn/NC@ZnO electrode material

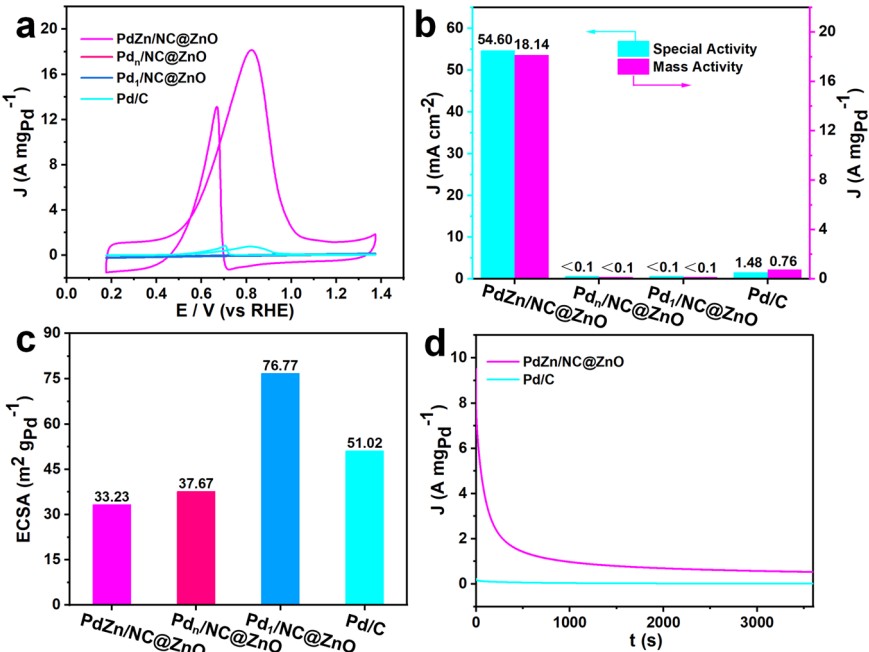

**Fig. 3 Catalytic activity and durability evaluation. a** CV curves of PdZn/NC@ZnO, Pd$_n$/NC@ZnO, Pd$_1$/NC@ZnO, and commercial Pd/C, recorded in N$_2$-saturated 1.0 M KOH and 1.0 M C$_2$H$_5$OH at room temperature at scan rate of 50 mV s$^{-1}$. **b** Mass and specific activities of PdZn/NC@ZnO and commercial Pd/C. **c** ECSA values of PdZn/NC@ZnO, Pd$_n$/NC@ZnO, Pd$_1$/NC@ZnO, and commercial Pd/C. **d** Chronoamperometry curves of PdZn/NC@ZnO and commercial Pd/C, recorded at their corresponding peak potentials at −0.30 V vs Ag/AgCl in aqueous solution containing 1.0 M KOH and 1.0 M C$_2$H$_5$OH.

(30.00 kW g$_{Pd}$$^{-1}$) is markedly improved in comparison to the Pd/C electrocatalyst (1.295 kW g$_{Pd}$$^{-1}$). Therefore, it can be concluded that the higher activity of PdZn/NC@ZnO at lower potentials as observed in the CV and chronoamperometry experiments still adapts to the real operational conditions of DEFC.

**Disclosing the superiority of Pd–Zn dual sites**. To better understand the superiority of Pd–Zn dual sites, we first probe into the reaction pathway of EOR over PdZn/NC@ZnO. The products in gas and liquid phase of the EOR reaction system are, respectively, analyzed by on-line GC, ion chromatography (IC), and $^1$H NMR (Supplementary Fig. 26). It is found out that acetate is only detected product in the EOR reaction over PdZn/NC@ZnO, and no CO$_2$ is formed. This demonstrates that the EOR proceeds the common reactive-intermediate pathway that ethanol is initially oxidized to acetaldehyde and subsequently to acetic acid or acetate in alkaline solution[34,45,61].

Density functional theory (DFT) calculations are further applied to analyze the EOR procedure on three different Pd-based active sites. For the surface of intermetallic PdZn model, we compare the surface energies of (001), (100) as well as (110) surfaces, which correspond to the (002), (200), and (110) surfaces observed on PdZn/NC@ZnO by AC HAADF-STEM images (Fig. 1g). The tendency of surface energy is (110) < (100) < (001) (Supplementary Table 4). Hence, the most stable (110) surface is selected as the reaction surface (Supplementary Fig. 27). Our theoretical studies suggested that for individual Pd sites the EOR process would be interrupted due to the exclusive hydrogen bond when CH$_3$CH$_2$O is formed. (Supplementary Figs. 28 and 29). As for Pd–Pd sites on Pd NPs, the ethanol adsorbs on Pd atoms and are prior to eliminate its methylene C–H bond during the EOR process, which agrees with previous studies (Fig. 4a and Supplementary Fig. 30)[62–65]. On the contrary, for Pd–Zn dual sites on PdZn NPs, present Zn atoms facilitate the adsorption of ethanol on them rather than on Pd atoms, and let the dehydrogenation of ethanol much easier to break O–H bond

initially instead of methylene C–H bond, adjusting the EOR procedure to a variant pathway (Fig. 4b and Supplementary Fig. 31). Free energy profiles of these two different pathways are described in Fig. 4c, d and Supplementary Fig. 32. It can be observed that the reaction procedure of initial dehydrogenation of ethanol (State 2) over Pd–Zn dual sites is easier than that over Pd–Pd sites (−37.88 vs −35.38 kcal mol$^{-1}$). After the 1st dehydrogenation step (State 2), CH$_3$CHOH and CH$_3$CH$_2$O are favorable on Pd (111) and PdZn (110) surface, respectively. The formed CH$_3$CH$_2$O could eliminate the possibility of parallel reactions to either CH$_3$CHO or CH$_3$COH in the 2nd dehydrogenation step (State3), which coexisted for the 2nd dehydrogenation step of CH$_3$CHOH on Pd (111) surface. Meanwhile, the desorption processes of acetic acid on the Pd–Zn dual sites are much easier than that on Pd–Pd sites (1.27 vs 7.72 kcal mol$^{-1}$). Therefore, Pd–Zn dual sites can enhance the interaction with ethanol and OH substrates and thus alter the EOR process into an energetically more favorable pathway, which contributes to the preeminent activity of PdZn/NC@ZnO in contrast to Pd$_1$/NC@ZnO, Pd$_n$/NC@ZnO, and commercial Pd/C.

## Discussion

In conclusion, we report that creating Pd–Zn dual sites with well exposure and homogeneity can boost the EOR to a large extent. Comparing with Pd$_n$/NC@ZnO with Pd–Pd sites and Pd$_1$/NC@ZnO with individual Pd sites, the PdZn/NC@ZnO with Pd–Zn dual sites displays the striking activity and good durability for EOR, even preceding commercial Pd/C significantly. DFT calculations manifest that Pd–Zn dual sites benefit the adsorption of ethanol and OH with respect to Pd–Pd sites and individual Pd sites, tuning the reaction pathway of EOR to proceed more smoothly with much lower energy barriers. Our study offers the clear comprehension for the role of alloyed dual sites on EOR, and may open up new opportunities to develop efficient electrocatalysts for DEFCs.

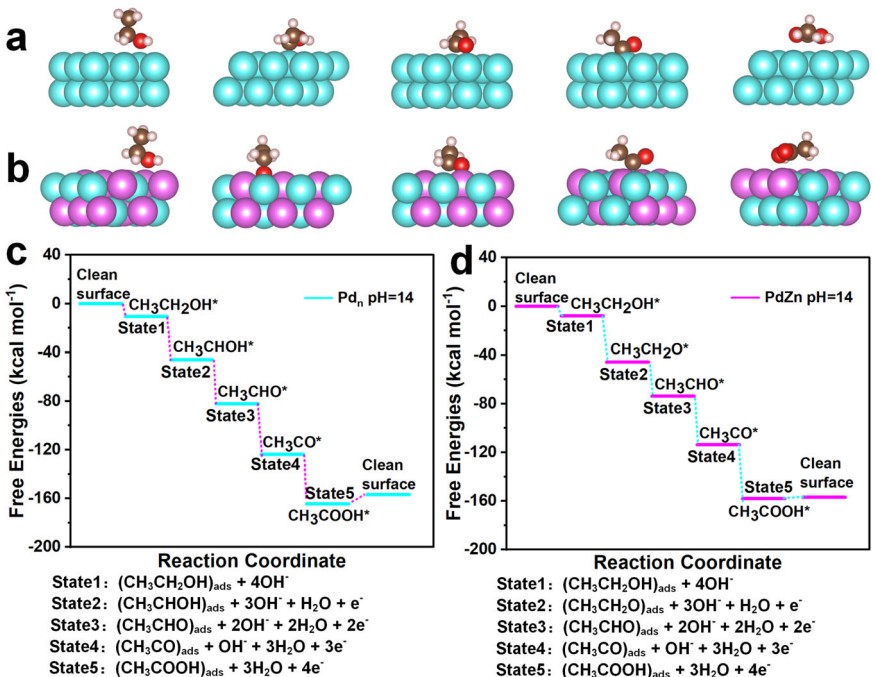

**Fig. 4 DFT calculated reaction procedure of EOR on Pd–Pd sites and Pd–Zn dual sites. a** DFT calculated models of Pd–Pd sites adsorbed with the reactive species from different EOR reaction states. **b** DFT calculated models of Pd–Zn dual sites adsorbed with the reactive species from different EOR reaction states. **c** DFT calculated free energy profiles of EOR over Pd–Pd sites (pH = 14, U = 0.82 V with respect to the RHE). **d** DFT calculated free energy profiles of EOR over Pd–Zn dual sites (pH = 14, U = 0.82 V with respect to the RHE).

## Methods

**Synthesis of ZnO@PDA**. ZnO nanorods were synthesized by a reported method[53]. Totally, 100.0 mg ZnO nanorods were dispersed in 20.0 mL Tris buffer solution (10.0 mM, pH = 8.5) by ultrasonication. Then dopamine hydrochloride solution (100.0 mg in 1.0 mL deionized water) was added to the ZnO suspension while stirring. The suspension was allowed to stir for 12 h at room temperature. The resulting products were washed by deionized water and ethanol three times and collected by centrifugation. After dried at 80 °C in an oven, the desired ZnO@PDA was obtained.

**Synthesis of Pd(OH)$_2$/ZnO@PDA**. A total of 22.5 mg of urea was added in 100 mL of deionized water. Then, 200.0 mg of ZnO@PDA was added into the solution and ultrasonicated for 30 min. Totally, 1.0 mL of 2.8 mg mL$^{-1}$ Na$_2$PdCl$_4$ aqueous solution was added and stirred for 3 h at room temperature. The mixture was then heated up to 90 °C for 12 h. The products were centrifuged and washed with deionized water three times. After drying in an oven at 80 °C overnight, Pd(OH)$_2$/ZnO@PDA was obtained.

**Synthesis of PdZn/NC@ZnO**. The obtained Pd(OH)$_2$/ZnO@PDA was treated at 800 °C in a home-made porcelain boat with a cap under H$_2$ (5% in Ar) atmosphere for 1 h in a tube furnace at a ramp rate of 5 °C min$^{-1}$ to provide PdZn/NC@ZnO.

**Performance comparison of different catalysts**. Electrochemical measurements were carried out with a three-electrode system on an electrochemical workstation (CHI 660, Shanghai Chenhua, China). Our experiments were performed with a saturated Ag/AgCl electrode (Ag/AgCl) electrode as the reference electrode. It was calibrated to E (RHE, reversible hydrogen electrode) from E(Ag/AgCl) by following the formula

$$E(\text{RHE}) = E(\text{Ag/AgCl}) + 0.197 + 0.05916\text{pH} \qquad (1)$$

10.0 mg catalysts were dispersed in 950.0 μL isopropanol, 950.0 μL deionized water, and 100.0 μL of 0.5 wt% Nafion solution and then followed by an intense sonication of 2 h to get a homogeneous 5.0 mg mL$^{-1}$ catalyst ink. After that, 20.0 μL inks (equal to 10.0 μg of catalyst) were deposited onto a 5.0 mm diameter glassy carbon rotating disk electrode (RDE, Pine Research Instrumentation) and dried to form a uniform thin film that was further used as a working electrode. Therefore, the actual Pd loadings of PdZn/NC@ZnO, Pd$_1$/NC@ZnO, Pd$_n$/NC@ZnO, and commercial Pd/C are 0.03008, 0.03458, 0.03983, and 0.50000 μg, respectively. A carbon rod was used as the counter electrode, and a saturated Ag/AgCl electrode was used as the reference electrode. All of the CV measurements were obtained at room temperature. The electrolyte solutions were purged with high-purity nitrogen for at

least 30 min before use. The working electrode was initially cycled between −0.85 and 0.35 V at 50.0 mV s$^{-1}$ in 1.0 M KOH for several cycles to remove the residual ligands on catalyst surface. Afterward, for the EOR measurement, the working electrodes were subject to CV scans between −0.85 and 0.65 V at 50 mV s$^{-1}$ in 1.0 M KOH and 1.0 M ethanol. The chronoamperometry measurements were conducted at −0.30 V in the solution of 1.0 M KOH and 1.0 M ethanol.

**Characterizations**. XRD patterns were recorded with a Rigaku D/max 2500Pc X-ray powder diffractometer with monochromatized Cu Kα radiation (λ = 1.5418 Å). TEM images were recorded by Hitachi-7700 working at 100 kV. HR-TEM, STEM, line scan analysis, and the corresponding EDX mapping were recorded by a JEOL JEM-2100F high-resolution TEM operating at 200 kV. AC HAADF-STEM images are taken on a JEOL JEMARM200F TEM/STEM with a spherical aberration corrector working at 300 kV. The lattice spacing is analyzed by Digital Micrograph software, with a calculation process explained as follows. The pixel intensities along each line parallel to the analyzed lattice fringes (several atoms length is usually included for high accuracy) are first added together, and then plotted vs. the distances along the perpendicular direction. The XPS spectrum was measured ex situ by a PHI Quantera SXM system under $3.1 \times 10^{-8}$ Pa using Al$^+$ radiation at room temperature. The binding energies were calibrated by referring C 1s peak to 284.8 eV. The metal content was determined by ICP-OES on Thermo Fisher IRIS Intrepid II. CO-DRIFTS characterizations are carried out on a Brucker Tenser II in situ infrared spectrometer with MCT detector using a home-made cell. The samples are pretreated with Ar at 25 °C for 1 h (20 mL min$^{-1}$), the background spectrum was collected. Then, CO gas was introduced into the sample holder for at 25 °C 1 h (20 mL min$^{-1}$) and spectra were collected every 5 min. The sample was flushing with Ar and spectra were collected every 5 min. XAS measurements were taken at BL14W1 station in Shanghai Synchrotron Radiation Facility (SSRF, operated at 3.5 GeV with a maximum current of 250 mA, Pd K-edge under fluorescence excitation mode). The XAS data of PdZn/NC@ZnO, Pd$_1$/NC@ZnO, and Pd$_n$/NC@ZnO samples were collected at room temperature in fluorescence excitation mode using a Lytle detector and Ru filter. Pd film and PdO were used as references and measured in a transmission mode using ionization chamber. The acquired XAS data were processed according to the standard procedures using the ATHENA module implemented in the IFEFFIT software packages. The XANES spectra were gained by subtracting the post-edge background from the overall absorption and then normalizing with respect to the edge-jump step. Subsequently, the χ(k) data were Fourier transformed to real (R) space using a hanning windows (dk = 1.0 Å$^{-1}$) to separate the EXAFS contributions from different coordination shells. Least-squares curve parameter fitting was carried out using the ARTEMIS module of IFEFFIT software packages to obtain the quantitative structural parameters around central atoms. Gas chromatography (GC, Shimadzu, Tracera (GC-2010 Plus A, Barrier Discharge Ionization Detector (BID)-2010 Plus))

with a GC column (Shinwa Chemical Industries, Micropacked ST) and He carrier gas (purity no less than 99.999%). Ion Chromatography (ICS-1100, Thermo Dionex) equipped with conductivity detector, AS-DV automatic sampler, protected and analytical column: AG19 (2*50 mm) and AS19 (4*250 mm) analytical column, eluent KOH concentration of 20 mM, flow rate of 1 mL min$^{-1}$, inhibition current of 50 mA, column temperature of 30 °C. $^{1}$H nuclear magnetic resonance (NMR) data were recorded with a Bruker Advance III (400 MHz) spectrometer. Fuel Cell Test System (Hephas 850e).

**DFT calculations**. All calculations in this work were performed by DMol$^{3}$ code[66]. The generalized gradient approximation with the Perdew–Burke–Ernzerhof functional[67] was selected to deal with the exchange and correlation function. For precisely treating the long-range van der Waals interactions, we employed the empirical correction in the Grimme scheme[68,69]. The DFT Semi-core Pseudopots (DSPP) and double numerical plus polarization (DNP) basis set were adopted for the Pd/Zn and C/H/O/N atoms, respectively. The convergence tolerance of energy and force of $1.0 \times 10^{-5}$ Ha, and $2.0 \times 10^{-3}$ Ha/Å during the fully geometry optimization. The vacuum space along z direction is set to be 15 Å to avoid interaction between the two neighboring images. A Monkhorst–pack mesh of $2 \times 2 \times 1$, $2 \times 1 \times 1$, and $1 \times 1 \times 1$ k-points was used in sampling the integrals over the Brillouin zone, for Pd–Pd sites, Pd–Zn dual sites, and individual Pd sites related models, respectively[70]. Calculated lattice constants for PdZn are 3.003 and 3.232 Å, which are in good agreement with them in Materialsprojects (2.895 and 3.342 Å)[71]. Slab models of $p$ $(3 \times 3)$ Pd (111) and $p$ $(2 \times 3)$ PdZn (110) were selected as the substrates for ethanol electro-oxidation reaction. Pd-N$_4$ (pyridine-N$_4$) embedded in $p$ $(6 \times 6)$ graphene model was selected as the single-atom catalyst substrate. The adsorption energies were calculated according to the formula,

$$E_{ads} = E(\text{adsorbate/sub}) - E(\text{adsorbate}) - E(\text{sub}) \quad (2)$$

where $E(\text{adsorbate/sub})$, $E(\text{adsorbate})$, and $E(\text{sub})$ represent the total energy of substrate with adsorbed species, the adsorbate species, and the clean substrate. The change in free energy for all the elementary steps is calculated based on the computational hydrogen electrode method developed by Norskov and his co-workers[72]. the reaction free energy $\Delta G$ is defined as the difference between free energies of the final and initial states and is given by the formula:

$$\Delta G = \Delta E + \Delta ZPE - T\Delta S + \Delta G_U + \Delta G_{pH} \quad (3)$$

where $\Delta E$ is the DFT calculated reaction energy of reactant and product molecules adsorbed on substeates; $\Delta ZPE$ and $\Delta S$ are the change in zero-point energies and entropy due to the reaction. The bias effect on the free energy of each initial, intermediate and final state involving an electron in the electrode is taken into accounts by shifting the energy of the state by

$$\Delta G_U = -neU \quad (4)$$

where $U$ is the electrode applied potential. $\Delta G_{pH}$ is the correction of the H$^+$ free energy at a pH different from 0:

$$\Delta G_{pH} = -k_B T ln[H^+] = pH \times k_B T ln10 \quad (5)$$

where $k_B$ is the Boltzmann constant and $T$ is the temperature. The equilibrium potential at pH = 14 was determined to be 0.402 V versus normal hydrogen electrode (NHE) according to the Nernst equation.

The considered EOR occurring in an alkaline electrolyte (pH = 14) are shown in the following steps[44,65,73]:

$$CH_3CH_2OH^* + OH^- \rightarrow CH_3CH_2O^* + H_2O + e^- \quad (1)$$

$$CH_3CH_2OH^* + OH^- \rightarrow CH_3CHOH^* + H_2O + e^- \quad (2)$$

$$CH_3CH_2OH^* + OH^- \rightarrow CH_3CHOH^* + H_2O + e^- \quad (3)$$

$$CH_3CHOH^* + OH^- \rightarrow CH_3CHO^* + H_2O + e^- \quad (4)$$

$$CH_3CHOH^* + OH^- \rightarrow CH_3CHO^* + H_2O + e^- \quad (5)$$

$$CH_3CHOH^* + OH^- \rightarrow CH_3CHO^* + H_2O + e^- \quad (6)$$

$$CH_3CHOH^* + OH^- \rightarrow CH_3CHO^* + H_2O + e^- \quad (7)$$

$$CH_3CO^* + OH^- \rightarrow CH_3COOH^* + e^- \quad (8)$$

## Data availability
The data supporting this study are available from the authors upon reasonable request.

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

## Acknowledgements

This work was supported by the National Key R&D Program of China (2018YFA0702003), the National Natural Science Foundation of China (21890383, 21871159, and 52002249), Science and Technology Key Project of Guangdong Province of China (2020B010188002), the Guangdong Basic and Applied Basic Research Foundation (2019A1515110025), the National Postdoctoral Program for Innovative Talents (BX20190167), the Shuimu Tsinghua Scholar Program, the China Postdoctoral Science Foundation (2020M670283). We thank the BL14W1 station of Shanghai Synchrotron Radiation Facility (SSRF) for XAFS measurements.

## Author contributions

Y.Q. performed the experiments, collected and analyzed the data, and wrote the paper. W.S. conducted the density functional theory calculation and analysis. Z.Z. helped with XANES and EXAFS spectrometry analyses. L.G. assisted in the AC HAADF-STEM characterization. J.J., J.S., Q.C., G.M., Q.X., Y.Z., H.T. and A.H. helped with data analysis and discussions. J.Z., D.W. and Y.L. conceived the experiments, planned synthesis, analyzed results, and wrote the paper.

## Competing interests

The authors declare no competing interests.
