## [Peer Review File · Nature Communications]

Construction of Pd-Zn Dual Sites to Enhance the Performance for Ethanol Electro-Oxidation ReactionReviewers' Comments:

Reviewer #1:

Remarks to the Author:

The authors of this manuscript present the synthesis and study of Pd-Zn Dual Site catalyst for ethanol electro-oxidation. Pd-Zn dual sites are shown to display 27 much higher activity for ethanol electro-oxidation, exceeding that of commercial Pd/C by a 28 factor of ~ 24 . Computational studies confirm the increased activity due to reduced energy barriers. The characterization of the materials is excellent, with very high quality HR-TEM and XAS studies. The activity in half cell is very high in terms of A per gram of Pd. Significantly higher than alternative catalysts reported in the literature. The authors demonstrate the advantage of the atomic level structure in enhancing the ethanol oxidation reaction.

The article is suitable for publication in Nature Communications after dealing with the following aspects. These include the need for new experimental data (major revisions).

The discussion on page 6 regarding the poisoning effect of carbonaceous species is incorrect. The forward peak in the CV are caused by the oxidation and passivation of Pd and the reverse peak is caused by the Pd reduction and reactivation of the Pd surface. The mechanism of direct oxidation to acetate means no poisoning intermediates are formed. This discussion must be corrected.

Regarding the proposed mechanism, the oxidation pathway passes through CH₃CO to form acetate as sole product. It is important to demonstrate the real nature of the oxidation products formed. This can be done through HPLC and GC analysis of the solution and headspace of the electrochemical cell after constant potential or current density experiments. Otherwise SNIFTIR spectroscopy could be used if available.

The stability to potential cycling is remarkable. Can the authors investigate the catalyst structure after testing? Is the atomic scale Pd-Zn structure maintained after stress testing? Pd is quite unstable to dissolution and agglomeration under potential cycling in alkaline media.

I am curious as to whether the enhanced performance and stability can be demonstrated in an actual alkaline membrane fuel cell. Such an excellent new anode catalyst would produce a very high power density. This would be an important addition to the article (see *Electrochemistry Communications*, 2009, 11(5), pp. 1077–1080, *ChemSusChem*, 2013, 6(3), pp. 518–528, *ChemSusChem*, 2009, 2(1), pp. 99–112)

Reviewer #2:

Remarks to the Author:

A sophisticated Pd-Zn/NC@ZnO catalyst is designed and synthesized in this work, which shows a decent EOR activity in alkaline solution. The importance of Pd-Zn dual site is highlighted, evidenced with XAS experiments and DFT calculation. However, the following points require more clarifications to make the conclusion exclusive.

Major:

1. The intermetallic structure of PdZn NPs is weakly evidenced by the STEM of a single NP (Figure 1 g): Considering the inhomogeneity of the NPs, providing more evidence like XRD or SAED can make the claim of the PdZn intermetallic alloy structure more convincing; Though it is mentioned in the main text (Page 4-5, line 76-78) the challenge to collect XRD signal of PdZn NPs, have the authors considered increasing the signals by other assistive approaches? For example, selectively etching ZnO away or increasing the PdZn loading on the support.

2. In Figure 2 a, both PdZn/NC@ZnO and Pd/NC@ZnO have a small peak at around 1.7 Å. In Table S1, the total coordination number of Pd in PdZn/NC@ZnO is smaller than 12. Is it possible that Pd-N or Pd-C compound also forms during the annealing process? With different percentages of these Pd-

N/C impurities in the catalysts, how will the catalytic property be affected?

3. Line 140-141, the XPS shift of PdZn/NC@ZnO is attributed to the electron transfer of Pd to Zn, which is against the electron-negativity trend (Pd 2.20 vs. Zn 1.65 from Wikipedia), can the author explain why?

4. The ECSA of PdZn/NC@ZnO is much higher than that of Pd_n/NC@ZnO and Pd₁/NC@ZnO. At the same time, both Pd_n/NC@ZnO and Pd₁/NC@ZnO show much lower activity than Pd/C. Does it indicate other factors except for Pd-Zn dual sites or Pd-Pd sites affect the catalytic property?

Minor:

1. Page 4, line 76, the size of PdZn NPs lacks standard deviation.
2. The captions of Figure 2 a, b, c, d do not match with the figures. Also, the tick label 2434.92 eV in the inset figure of 2b is not right, and Pd foil signal is not shown.
3. Page S4, the description "hydrogen adsorption charge" doesn't match with the equation. In addition, when using CO stripping method to determine the ECSA, q_{CO} should be $420 \mu C/cm^2$. (Lukaszewski, M.; Soszko, M.; Czerwiński, A. Int. J. Electrochem. Sci. 2016, 11 (6), 4442–4469.)
4. Figure S6, wrong crystal structure of intermetallic PdZn.

Reviewer #3:

Remarks to the Author:

In this article, the authors report the synthesis of a PdZn catalyst that is tested in the electrooxidation of ethanol aiming at ethanol fuel cell. The reported activity is high and the study is sustained by an extensive characterization and DFT computations. The discussion of the results is minimalist.

A major issue in using ethanol fuel cell is the possibility of performing the complete oxidation towards CO₂. How does this catalyst behave regarding this aspect?

Reporting the activity per mass of Pd is not the best way to compare the activity of exposed catalytic site. Ideally, this should be reported per surface atom.

In Figure 3, the caption has mixed up c. and d. At which potential is the chronoamperometry performed?

I have several concerns about the computations.

All computational details are only provided in SI making the reading frustrating. Dispersion is missing (and could affect the results with an ethanol molecule that is binding essentially through van der Waals interactions with the metallic surface). The choice of the PdZn model could be better related to the characterization. It seems different in the main and in the supplementary material !

For isolated Pd, the authors should check if OH and EtOH can adsorb on the same site. Currently, the structures that are presented are not convincing. Besides, a Eley-Rideal mechanism cannot be so easily discarded. This is clearly a weakness of the study.

The authors claim that they use the CHE method to take into account the potential. But they don't provide the potential at which they computed the reaction profile. And clearly, this is not a potential at which the reaction can proceed since the overall reaction to acetic acid is endothermic in Figure 4. This needs absolutely to be corrected to show results that can indeed be discussed in regard of the experimental results. Besides, the caption of Figure 4 is not coherent with the approach since the authors consider the basic medium in their state but underline the dissociation of a H⁺ + e⁻ pair in the

caption.

Last, the barriers provided for the CH and OH scission on Pd are much higher than the ones found in the literature (see for instance Neurock work on ethanol oxidation, *Journal of Catalysis* Volume 299, March 2013, Pages 261-271). Proposing barriers greater than 150kJ/mol is clearly unrealistic. I doubt the quality of those results and thus of the overall conclusion about the Zn effect! In figure 4, the TS energy should be also shown in the profiles.

Reviewer #1:

The authors of this manuscript present the synthesis and study of Pd-Zn Dual Site catalyst for ethanol electro-oxidation. Pd-Zn dual sites are shown to display 27 much higher activity for ethanol electro-oxidation, exceeding that of commercial Pd/C by a 28 factor of ~24. Computational studies confirm the increased activity due to reduced energy barriers. The characterization of the materials is excellent, with very high quality HR-TEM and XAS studies. The activity in half cell is very high in terms of A per gram of Pd. Significantly higher than alternative catalysts reported in the literature. The authors demonstrate the advantage of the atomic level structure in enhancing the ethanol oxidation reaction.

The article is suitable for publication in Nature Communications after dealing with the following aspects. These include the need for new experimental data (major revisions).

Comment 1. *The discussion on page 6 regarding the poisoning effect of carbonaceous species is incorrect. The forward peak in the CV are caused by the oxidation and passivation of Pd and the reverse peak is caused by the Pd reduction and reactivation of the Pd surface. The mechanism of direct oxidation to acetate means no poisoning intermediates are formed. This discussion must be corrected.*

Response: We sincerely appreciate your valuable suggestion. After referring to many literatures on electrocatalytic ethanol oxidation in alkaline media, it is accepted that the EOR in alkaline medium can proceed in two reaction pathways, the reactive-intermediate pathway and the poisoning-intermediate pathway (*Nano Energy*, **77**, 105116 (2020); *J. Am. Chem. Soc.* **133**, 15172-15183 (2011); *ChemSusChem* **2**, 99 – 112 (2009); *J. Am. Chem. Soc.* **142**, 13833-13838 (2020); *Nat. Commun.* **8**, 14136 (2017)). In this work, we have affirmed the final product of EOR is only acetic acid instead of CO₂ (for details, see Supplementary Fig. 26). Therefore, ethanol is first oxidized to acetaldehyde and subsequently to acetic acid or acetate in alkaline solution without the C–C bond being broken, which accords with the reactive-intermediate pathway and indeed no toxic substances are produced (*Nano Energy*, **77**, 105116 (2020); *J. Am. Chem. Soc.* **133**, 15172-15183 (2011)); *ChemSusChem* **2**, 99 – 112 (2009)).

Although current density ratio of forward scan (J_f) to backward scan (J_b) cannot show the stability of the catalyst, we have added more characterizations on the catalyst after potential cycles in the supplementary information, which can reflect its structural stability. We detect the content of residual metal after EOR reaction in PdZn/NC@ZnO and the alkaline solution by ICP-OES. It is found that Pd (< 100 $\mu\text{g/mL}$) and Zn (< 20 $\mu\text{g/mL}$) are nearly not detected in the reaction solution, and the content of Pd in PdZn/NC@ZnO keeps the same as 0.3 wt% after the stability test (Table S3). As revealed by STEM and EDS line scanning measurements, PdZn/NC@ZnO after the stability test shows no observable changes in the morphology and composition (Supplementary Fig. 22). XPS spectra of PdZn/NC@ZnO after the stability test manifest the unchanged Pd 3d signals (Supplementary Fig. 23). Further AC HAADF-STEM characterizations also verify the intermetallic PdZn (P4/mmm) crystal structure of PdZn/NC@ZnO after the stability test (Supplementary Fig. 24). These results can prove the desired stability of PdZn/NC@ZnO for EOR, and we have revised the manuscript accordingly as follows:

p. 10 “Barely no residual Pd and Zn are detected in the reaction mixture, and the content of Pd in PdZn/NC@ZnO keeps the same as 0.3 wt% after the stability test (Table S3). Corresponding characterizations of PdZn/NC@ZnO after potential cycles show no observable changes in the morphology and composition (Supplementary Figs. 22-24). These results prove the desired stability of PdZn/NC@ZnO for EOR.”

Table S3. ICP-OES detections of PdZn/NC@ZnO and solution after the stability test.

Element	PdZn/NC@ZnO (wt%)	Solution ($\mu\text{g/mL}$)
Pd	0.3	< 100
Zn	-	< 20

Supplementary Figure 22 | STEM image and EDS line scanning. **a** STEM image of PdZn/NC@ZnO after the stability test, the inset in (a) is the size distribution histogram of PdZn NPs. Scale bar 100 nm. **b** EDS line scanning of PdZn/NC@ZnO after the stability test.

Supplementary Figure 23 | Pd 3d XPS spectra of PdZn/NC@ZnO before and after the stability test.

Supplementary Figure 24 | AC HAADF-STEM characterization of PdZn/NC@ZnO after the stability test. **a** AC HAADF-STEM image of one PdZn nanoparticle in PdZn/NC@ZnO after the stability test. Scale bar, 2.5 nm. **b** AC HAADF-STEM elemental mappings of one PdZn nanoparticle in PdZn/NC@ZnO after the stability test. Scale bar, 5 nm.

Comment 2. Regarding the proposed mechanism, the oxidation pathway passes through CH_3CO

to form acetate as sole product. It is important to demonstrate the real nature of the oxidation products formed. This can be done through HPLC and GC analysis of the solution and headspace of the electrochemical cell after constant potential or current density experiments. Otherwise SNIFTIR spectroscopy could be used if available.

Response: We appreciate you for this insightful and constructive recommendation on verifying the real nature of the oxidation products formed. Thus, we have affirmed the final product of EOR is acetic acid only instead of CO₂ (Supplementary Fig. 26). Therefore, ethanol is first oxidized to acetaldehyde and subsequently to acetic acid or acetate in alkaline solution without breaking C–C bonds, which accords with the reactive-intermediate pathway (*Nano Energy*, **77**, 105116 (2020); *J. Am. Chem. Soc.* **133**, 15172-15183 (2011); *ChemSusChem* **2**, 99 – 112 (2009)). Based on these results, we have added the identification of the electrooxidation products of ethanol in the revised manuscript and supplementary information as follows:

p.11 “To better understand the superiority of Pd-Zn dual sites, we firstly probe into the reaction pathway of EOR over PdZn/NC@ZnO. The products in gas and liquid phase of the EOR reaction system are respectively analyzed by on-line GC, ion chromatography (IC) and ¹H NMR (Supplementary Fig. 26). It is found out that acetate is only detected products in the EOR reaction over PdZn/NC@ZnO, and no CO₂ is formed. This demonstrates that the EOR proceeds the common reactive-intermediate pathway that ethanol is initially oxidized to acetaldehyde and subsequently to acetic acid or acetate in alkaline solution^{34, 45, 62}.”

Supplementary Figure 26 | Detections on the products in gas and liquid phase of EOR over PdZn/NC@ZnO. **a** On-line GC spectrum of the gas phase in EOR. **b** IC spectrum of the liquid phase in EOR **c** ^1H NMR spectrum of the liquid phase in EOR.

Comment 3. *The stability to potential cycling is remarkable. Can the authors investigate the catalyst structure after testing? Is the atomic scale Pd-Zn structure maintained after stress testing? Pd is quite unstable to dissolution and agglomeration under potential cycling in alkaline media.*

Response: We appreciate you for this insightful and constructive recommendation on the structure stability of Pd-Zn. To investigate the catalyst structure after testing, STEM, EDS line scanning, XPS, and AC HAADF-STEM analyses are carried out respectively on PdZn/NC@ZnO after the stability test. As revealed by STEM and EDS line scanning analysis, the catalyst after the stability test shows no observable changes in the morphology and composition (Supplementary Fig. 22). It can be evidenced from XPS spectra that Pd 3d signals of PdZn/NC@ZnO have no changes before and after the stability test (Supplementary Fig. 23). Atomic-resolution EDS elemental mapping analysis manifests that Pd and Zn still homogeneously disperse over the entire nanoparticle after the stability test (Supplementary Fig. 24a). The typical AC-HAADF STEM image of one nanoparticle in PdZn/NC@ZnO after the stability test also exhibits clear bright dots whose ordering accords with the atomic arrangement of (002) plane in the intermetallic PdZn (P4/mmm) crystal structure (Supplementary Fig. 24b). These results testify the remained intermetallic nature of PdZn NPs with uniform Pd-Zn dual sites in PdZn/NC@ZnO after the stability test.

Although Pd is quite unstable to dissolution and agglomeration under potential cycling in alkaline media, alloying Pd with other metals like Sn, Ge and Zn can improve its stability under alkaline conditions (*Angew. Chem. Int. Ed.* **55**, 9030-9035 (2016); *J. Catal.* **382**, 181-191 (2020); *ACS Catal.* **10**, 1171-1184 (2019)). We detect the content of residual metal after EOR reaction in PdZn/NC@ZnO and the alkaline solution by ICP-OES. Finally, Pd (< 100 ug/mL) and Zn (< 20 ug/mL) are nearly not detected in the reaction solution, and the content of Pd in PdZn/NC@ZnO keeps the same as 0.3 wt% after the stability test (Table S3). These results convinced the stability of our prepared PdZn/NC@ZnO for EOR in alkaline media.

To elucidate the stability of PdZn/NC@ZnO for EOR in more detail, we have revised the manuscript as follows:

p. 10 “Barely no residual Pd and Zn are detected in the reaction mixture, and the content of Pd in PdZn/NC@ZnO keeps the same as 0.3 wt% after the stability test (Table S3). Corresponding characterizations of PdZn/NC@ZnO after potential cycles show no observable changes in the morphology and composition (Supplementary Figs. 22-24). These results prove the desired stability of PdZn/NC@ZnO for EOR.”

Table S3. ICP-OES detections of PdZn/NC@ZnO and solution after the stability test.

Element	PdZn/NC@ZnO (wt%)	Solution ($\mu\text{g/mL}$)
Pd	0.3	< 100
Zn	-	< 20

Supplementary Figure 22 | STEM image and EDS line scanning. **a** STEM image of PdZn/NC@ZnO after the stability test, the inset in (a) is the size distribution histogram of PdZn NPs. Scale bar 100 nm. **b** EDS line scanning of PdZn/NC@ZnO after the stability test.

Supplementary Figure 23 | Pd 3d XPS spectra of PdZn/NC@ZnO before and after the stability test.

Supplementary Figure 24 | AC HAADF-STEM characterization of PdZn/NC@ZnO after the stability test. a AC HAADF-STEM image of one PdZn nanoparticle in PdZn/NC@ZnO after the stability test. Scale bar, 2.5 nm. **b** AC HAADF-STEM elemental mappings of one PdZn nanoparticle in PdZn/NC@ZnO after the stability test. Scale bar, 5 nm.

Comment 4. I am curious as to whether the enhanced performance and stability can be demonstrated in an actual alkaline membrane fuel cell. Such an excellent new anode catalyst would produce a very high power density. This would be an important addition to the article (see Electrochemistry Communications, 2009, 11(5), pp. 1077–1080, ChemSusChem, 2013, 6(3), pp. 518–528, ChemSusChem, 2009, 2(1), pp. 99–112)

Response: Thank you very much for this insightful and constructive recommendation. As you recommended, we have evaluated the performance of PdZn/NC@ZnO in the actual alkaline membrane fuel cell compared with commercial Pd/C. As shown in Supplementary Fig. 25, the polarization and power density curves in single DEFC shows that the open circuit voltage of the fuel cell containing PdZn/NC@ZnO and Pd/C as anodes is 0.846 and 0.843 V respectively. The DEFC performances of PdZn/NC@ZnO (72.30 mW cm^{-2}) are better than that of Pd/C (64.75 mW cm^{-2}). Notably, when normalizing the power density values of the DEFC with respect to the Pd loading, the electrical performance of PdZn/NC@ZnO electrode material ($30.00 \text{ kW/g}_{\text{Pd}}$) is markedly improved in comparison to the Pd/C electrocatalyst ($1.295 \text{ kW/g}_{\text{Pd}}$). On the basis of these results, we believe that the higher activity of PdZn/NC@ZnO at lower potentials as observed in the CV and chronoamperometry experiments still adapts to the real operational conditions of DEFC. To demonstrate the performance and stability of PdZn/NC@ZnO in an actual alkaline membrane fuel cell, the manuscript and supplementary information have been revised accordingly as follows:

p. 10 “The performance of PdZn/NC@ZnO for EOR are further evaluated in an actual alkaline membrane fuel cell. The polarization and power density curves in single DEFC with different

electrocatalysts as anodes are shown in Supplementary Fig. 25. The open circuit voltage of the fuel cell containing PdZn/NC@ZnO and Pd/C electrocatalysts is 0.846 and 0.843 V respectively. The DEFC performances of PdZn/NC@ZnO (72.30 mW cm^{-2}) are better than that of Pd/C (64.75 mW cm^{-2}). Notably, when normalizing the power density values of the DEFC with respect to the Pd loading, the electrical performance of PdZn/NC@ZnO electrode material ($30.00 \text{ kW/g}_{\text{Pd}}$) is markedly improved in comparison to the Pd/C electrocatalyst ($1.295 \text{ kW/g}_{\text{Pd}}$). Therefore, it can be concluded that the higher activity of PdZn/NC@ZnO at lower potentials as observed in the CV and chronoamperometry experiments still adapts to the real operational conditions of DEFC.”

Supplementary Figure 25 | The polarization and power density curves for DEFC. PdZn/NC@ZnO (0.8 mg/cm^2 total catalyst loading) and Pd/C (1.0 mg/cm^2 total catalyst loading) are used as anode catalysts respectively. All experiments are done using 1.0 M ethanol in 1.0 M KOH as a fuel feed. The fuel cell is conditioned at $50 \text{ }^\circ\text{C}$ with 2.0 mL/min fuel and 100 sccm O_2 . Electrolyte: Nafion212 membrane.

Reviewer #2:

A sophisticated Pd-Zn/NC@ZnO catalyst is designed and synthesized in this work, which shows a decent EOR activity in alkaline solution. The importance of Pd-Zn dual site is highlighted, evidenced with XAS experiments and DFT calculation. However, the following points require more clarifications to make the conclusion exclusive.

***Comment 1.** The intermetallic structure of PdZn NPs is weakly evidenced by the STEM of a single NP (Figure 1 g): Considering the inhomogeneity of the NPs, providing more evidence like XRD or SAED can make the claim of the PdZn intermetallic alloy structure more convincing; Though it is mentioned in the main text (Page 4-5, line 76-78) the challenge to collect XRD signal of PdZn NPs, have the authors considered increasing the signals by other assistive approaches? For example, selectively etching ZnO away or increasing the PdZn loading on the support.*

Response: Thanks for your suggestion. As you suggested, in order to further convince the intermetallic structure of PdZn NPs, we try to raise the loading of Pd to increase the amount and particle size of PdZn NPs and etch ZnO away in the final PdZn/NC@ZnO sample for enhancing the XRD signal of PdZn NPs. By this way, the featured intermetallic PdZn signals can be expectedly evidenced in the XRD patterns (Supplementary Fig. 5c), which further affirms the nature of PdZn NPs in PdZn/NC@ZnO being intermetallic PdZn compounds. Thus, to make the structure identification of PdZn/NC@ZnO more convinced, we have revised the manuscript and supplementary information as follows:

p. 4-5. “After the ZnO being etched away, there are still PdZn NPs on the NC carrier in the Supplementary Fig. 5a, and only graphitic carbon peaks are observed in the XRD patterns because of the PdZn NPs being too small (Supplementary Fig. 5c). When the loading of Pd precursor is increased five times, it can be seen that PdZn NPs become larger and exhibit featured intermetallic PdZn signals in the XRD patterns (Supplementary Fig. 5c), which suggests the nature of the PdZn NPs in PdZn/NC@ZnO being PdZn intermetallic compounds^{55,56}.”

Supplementary Figure 5 | Characterizations of PdZn/NC and PdZn/NC@ZnO (5%). **a** TEM image of PdZn/NC. Scale bar 100 nm. **b** TEM image of PdZn/NC@ZnO (5%). Scale bar 200 nm. **c** XRD patterns of PdZn/NC (black line) and PdZn/NC@ZnO (5%) (blue line).

Comment 2. In Figure 2 a, both PdZn/NC@ZnO and Pd_n/NC@ZnO have a small peak at around 1.7 Å. In Table S1, the total coordination number of Pd in PdZn/NC@ZnO is smaller than 12. Is it possible that Pd-N or Pd-C compound also forms during the annealing process? With different percentages of these Pd-N/C impurities in the catalysts, how will the catalytic property be affected?

Response: Thanks a lot for this insightful and constructive recommendation on the FT-EXAFS at the Pd k-edge of the PdZn/NC@ZnO and Pd_n/NC@Zn. In our fitting model, the first shell of PdZn

structure, it shows 5-coordinated Pd-Zn with 2.674 Å path length and the second shell exhibits 2-coordinated Pd-Pd with 2.914 Å. These evidences prove that the coordination number of Pd-Zn is less than 12. It has been observed that metal nanoparticles (NPs) can be encapsulated by ultrathin overlayers derived from the supports under certain treatment or reaction conditions, which is the main characteristic of the classical strong metal–support interaction (SMSI) effect (*Appl. Catal. B.* **263**, 118355 (2020); *Nat. Commun.* **11**, 3220 (2020); *J. Mater. Chem.* **19**, 5934-5939 (2009); *J. Am. Chem. Soc.* **142**, 17167-17174 (2020); *ACS Appl. Mater. Inter.* **12**, 31467-31476 (2020)). Pd in the surface particles will interact with the carrier by SMSI effect. It causes multiple scattering peaks in XAFS, but the proportion is relatively low. XAFS represents the statistical average, Pd-C and Pd-N bonds may exist in PdZn/NC@ZnO, but the dominant structure fit is intermetallic PdZn compound (*Chem. Sci.*, **10**, 8292-8298 (2019); *Acc. Chem. Res.* **54**, 11, 2660–2669 (2021); *ACS Catal.* **10**, 2231–2259 (2020); *Adv. Mater.* **31**, 1900509 (2019)). With different percentages of these Pd-N/C impurities in the catalysts, we think it will not affect the catalytic property obviously.

Comment 3. Line 140-141, the XPS shift of PdZn/NC@ZnO is attributed to the electron transfer of Pd to Zn, which is against the electron-negativity trend (Pd 2.20 vs. Zn 1.65 from Wikipedia), can the author explain why?

Response: Thank you very much for this question. In metallic zinc ($3d^{10}4s^{2-x}4p^x$ electronic configuration (*Phys. Rev. B.* **22**, 4604 (1980); *Surf. Sci.* **131**, L390-L398 (1983)) the 3d band is very stable, appearing at ~ 10 eV below the Fermi level (*Phys. Rev. B* **22**, 4604 (1980); *Surf. Sci.* **131**, L390-L398 (1983); *Phys. Rev. B* **8**, 2392 (1973)). This fact makes the d-d bonding interactions negligible in systems that contain zinc and a transition metal.

In general, the chemical bonds between Zn and transition metals are considerably weaker than the typical bonds between two transition metals (*Chem. Phys.* **97**, 9427-9439 (1992)). In spite of this, zinc is able to induce substantial changes in the properties of a transition metal (*Chem. Rev.* **75**, 291-305 (1975); *J. Am. Chem. Soc.* **107**, 7216-7218 (1985); *J. Catal.* **116**, 361-372 (1989); *Surf. Sci.* **209**, 77-88 (1989); *Phys. Chem.* **98**, 5758-5764 (1994)). Zinc has a 4s valence band that is almost fully occupied, while the 4p band is almost empty (*Phys. Rev. B* **22**, 4604 (1980); *Surf. Sci.* **131**, L390-L398 (1983)). Thus, depending on the electronegativity of the transition metal, zinc can behave as an electron donor or an electron acceptor when present in a bimetallic surface (*Phys. Chem.* **100**, 381–389 (1996)). For example, in PdZn surfaces (*Phys. Chem.* **98**, 5758-5764 (1994)), the Pd atoms exhibit electronic and chemical perturbations that are as large as those found for Pd bonded to Zn and much bigger than those seen when Pd is bonded to late-transition metals. The formation of Pd-Zn bonds produces a large depletion in the density of Pd 4d states around the Fermi level and a positive binding energy shift of ~ 1 eV in the centroid of the Pd 4d band (*Phys. Chem.* **98**, 5758-5764 (1994); *Surf. Sci.* **120**, 239-250 (1982)). The Pd 3d region presented a doublet of Pd $3d_{5/2}$ and Pd $3d_{3/2}$ at 336.35 and 341.75 eV can be assigned to Pd in PdZn alloy. With accordance to the literature, the bimetallic bonding with Zn produces positive binding energy (BE) shift in the core levels and valence d band of the group 10 metals (*J. Phys. Chem.* **100**, 381–

389(1996)). The positive shift is connected with the reduction of electron population and subsequent shift of the valence d orbital (*Appl. Catal. B*, **154**, 316-328 (2014)).

Comment 4. *The ECSA of PdZn/NC@ZnO is much higher than that of Pd_n/NC@ZnO and Pd₁/NC@ZnO. At the same time, both Pd_n/NC@ZnO and Pd₁/NC@ZnO show much lower activity than Pd/C. Does it indicate other factors except for Pd-Zn dual sites or Pd-Pd sites affect the catalytic property?*

Response: Thanks for your question. After carefully checking our calculation procedure of ECSA values, we have to apology for our carelessness in the processing method of CO stripping data because of the linear background correcting being left out. For this reason, the area of integration is much larger which is wrong and lead to the much higher ECSA value of PdZn/NC@ZnO in contrast to Pd_n/NC@ZnO and Pd₁/NC@ZnO. Therefore, we corrected the data processing method with the the linear background correcting and only integrated the area of CO oxidation peak. ECSA was determined by integrating the charge of the CO_{ad} layer involved in an anodic stripping peak as below:

$$ECSA_{CO} = \frac{Q_{CO}}{m \cdot q_{CO}}$$

where Q_{CO} represents the Coulombic charge of the CO_{ad} layer (μC) involved in an anodic stripping peak, m is the mass of Pd on electrode surface, q_{CO} (420 μC/cm²) is the charge required for electrochemical oxidation of monolayers of adsorbed CO (*Appl. Catal. B*, **263**, 118304 (2020)).

We repeated the calculation ECSA with Q_{CO}, and the results are shown in Fig. 3c in the revised manuscript. After correction, Pd₁/NC@ZnO showed the largest ECSA of 76.77 m²/g_{Pd}, and PdZn/NC@ZnO, Pd_n/NC@ZnO, and commercial Pd/C display the ECSA as 33.23, 37.67, and 51.02 m²/g_{Pd}, respectively. We also calculate the specific activity again with corrected ECSA values, and the results are shown in Fig. 3b in the revised manuscript. Owing to the smaller ECSA corrected, the intrinsic activity of PdZn/NC@ZnO reaches up to 37 times higher than that of Pd/C. Therefore, we believe the significant improved activity of PdZn/NC@ZnO derives from the distinctive Pd-Zn dual sites being superior to Pd-Pd sites rather than other factors like ECSA of catalysts. Correspondingly, we have modified the manuscript to describe these results as follows:

p. 9-10. “The specific activity of PdZn/NC@ZnO is 54.60 mA/cm², also surpassing that of Pd/C (1.48 mA/cm²) by ~37 times (Fig. 3b). Electrochemical surface area (ECSA) for PdZn/NC@ZnO is determined to be 33.23 m²/g_{Pd}, smaller than that for Pd_n/NC@ZnO (36.76 m²/g_{Pd}), Pd₁/NC@ZnO (76.77 m²/g_{Pd}) and Pd/C (51.02 m²/g_{Pd}) (Fig. 3c and Supplementary Fig. 20)⁶¹. This implies the great intrinsic activity of active Pd-Zn sites in PdZn/NC@ZnO for EOR.”

Figure 3 | Catalytic activity and durability evaluation. **b** Mass and specific activities of PdZn/NC@ZnO and commercial Pd/C. **c** ECSA values of PdZn/NC@ZnO, Pd_n/NC@ZnO, Pd₁/NC@ZnO, and commercial Pd/C.

Supplementary Figure 20 | CO stripping plots for determining ECSA. **a** CO stripping plot of PdZn/NC@ZnO between 0.10 and 1.40 V vs RHE in 1.0 M KOH at a sweep rate of 0.02 V/s. **b** CO stripping plot of commercial Pd/C between 0.10 and 1.40 V vs RHE in 1.0 M KOH at a sweep rate of 0.02 V/s.

Comment 5. Page 4, line 76, the size of PdZn NPs lacks standard deviation.

Response: Thank you very much for this suggestion. We have calculated the standard deviation of the size distribution histogram and added to Fig. 1d, 1h and Supplementary Fig. 22a. To make the description more accurate, we have revised the manuscript and supplementary information as below:

p. 4 “The statistical distribution reveals the average particle size of such PdZn NPs being as small as 4.75 ± 0.77 nm (the inset in Fig. 1d and Supplementary Fig. 3).”

p. 6 “In contrast, when treated at low temperature (400 °C) to avoid the Zn vapor, only smaller Pd NPs (2.75 ± 0.66 nm) with Pd-Pd sites are formed in the catalyst (Pd_n/NC@ZnO) as deduced from STEM, TEM and EDS line scanning tests (Fig. 1h, Supplementary Figs. 9-12).”

Figure 1 | Synthetic scheme and characterizations of synthesized catalysts. **d** STEM image of PdZn/NC@ZnO, the inset is the size distribution histogram of PdZn nanoparticles in PdZn/NC@ZnO. Scale bar, 50 nm. **h** STEM image of Pd_n/NC@ZnO, the inset is the size distribution histogram of Pd nanoparticles in Pd_n/NC@ZnO. Scale bar, 20 nm.

Supplementary Figure 22 | STEM image and EDS line scanning. **a** STEM image of PdZn/NC@ZnO after the stability test, the inset in (a) is the size distribution histogram of PdZn NPs. Scale bar 100 nm.

Comment 6. The captions of Figure 2 a, b, c, d do not match with the figures. Also, the tick label 2434.92 eV in the inset figure of 2b is not right, and Pd foil signal is not shown.

Response: Thank you very much for careful reading. We feel sorry for our carelessness, we have carefully checked our manuscript. The graph annotation has been corrected in the revised manuscript. As for the Pd foil signal in the inset figure of Fig. 2b, it actually overlaps with the Pd_n/NC@ZnO signal and so cannot be seen from the figure because of Pd component in

Pd_n/NC@ZnO exists as pure metallic state which is consistent with Pd foil.

Figure 2 | X-ray absorption spectroscopy characterization of the catalysts. a FT-EXAFS spectra at the Pd k-edge of PdZn/NC@ZnO, Pd_n/NC@Zn and Pd₁/NC@ZnO with Pd foil as the reference. **b** XANES spectra at the Pd k-edge of PdZn/NC@ZnO, Pd_n/NC@Zn and Pd₁/NC@ZnO with Pd foil as the reference. **c** XPS spectra in Pd 3d region of PdZn/NC@ZnO, Pd_n/NC@Zn and Pd₁/NC@ZnO. **d** DRIFT spectra for PdZn/NC@ZnO, Pd_n/NC@Zn and Pd₁/NC@ZnO.

*Comment 7. Page S4, the description “hydrogen adsorption charge” doesn’t match with the equation. In addition, when using CO stripping method to determine the ECSA, q_{CO} should be $420 \mu C/cm^2$. (Lukaszewski, M.; Soszko, M.; Czerwiński, A. *Int. J. Electrochem. Sci.* 2016, 11 (6), 4442–4469.)*

Response: Thank you very much for careful reading. We feel sorry for our carelessness, we have carefully checked our manuscript. ECSA is determined again by integrating the charge of the CO_{ad} layer involved in an anodic stripping peak as below:

$$ECSA_{CO} = \frac{Q_{CO}}{m \cdot q_{CO}}$$

where Q_{CO} represents the Coulombic charge of the CO_{ad} layer (μC) involved in an anodic stripping peak, m is the mass of Pd on electrode surface, q_{CO} ($420 \mu C/cm^2$) is the charge required

for electrochemical oxidation of monolayers of adsorbed CO. (*Appl. Catal. B.* **263**, 118304 (2020); *J. Mater. Chem. A* **4**, 7950-7961 (2016); *Carbon* **149**, 370-379 (2019); *J. Catal.* **381**, 316-328 (2020); *Electrochem. Sci.* **11**, 4442–4469 (2016)).

The description of ECSA has been corrected in the revised supplementary information as below:

p. S4 “For CO stripping tests, CO oxidation experiments were carried out in the solution of 1.0 M KOH 20 mV s^{-1} . Before the test, the solution was purged with N_2 for 30 min and then was bubbled with CO gas (99.9%) for 15 min at 0.1 V to achieve the maximum coverage of CO at the Pd active centres. The residual CO in the solution was excluded by nitrogen for 30 min. The ECSA were obtained by integrating the charge of the COad layer involved in an anodic stripping peak between 0.10 V and 1.40 V vs RHE after the linear background correctting. The ECSA of Pd can be calculated based on the following equation:

$$\text{ECSA}_{\text{CO}} = \frac{Q_{\text{CO}}}{m \cdot q_{\text{CO}}}$$

where Q_{CO} is the charge for the oxidation of CO (mC/cm^2), is m the loading amount of metal, and q_{CO} is the charge required to oxidize the monolayer of CO ($420 \mu\text{C}/\text{cm}^2$) on the catalyst.”

Comment 8. *Supplementary Fig. 6, wrong crystal structure of intermetallic PdZn.*

Response: Thanks for your suggestion. The structure in the Supplementary Fig. 6 (Supplementary Fig. 7 in the revised supplementary information) is actually the unit cell structure of standard intermetallic PdZn compound (P4/mmm). To make the picture and description more clearly and accurately, we present the atomic arrangement corresponding to the (002) plane in the Supplementary Fig.7 instead of the whole unit cell structure, which matches the observed crystal plane on the PdZn NP.

Supplementary Figure 7 | AC HAADF-STEM image of a PdZn NP. **a** AC HAADF-STEM image of a PdZn NP in bright field. Scale bar 5 nm. **b** AC HAADF-STEM image of a PdZn NP in dark field, the insert is crystal structure of PdZn (002) Plane, Pd atoms (blue) and Zn atoms (purple). Scale bar 5 nm.

Reviewer #3:

In this article, the authors report the synthesis of a PdZn catalyst that is tested in the electrooxidation of ethanol aiming at ethanol fuel cell. The reported activity is high and the study is sustained by an extensive characterization and DFT computations. The discussion of the results is minimalist.

Response:

***Comment 1.** A major issue in using ethanol fuel cell is the possibility of performing the complete oxidation towards CO₂. How does this catalyst behave regarding this aspect?*

Response: Thanks for your question. To distinguish whether ethanol can be completely oxidized to CO₂ by PdZn/CN@ZnO, we analyze the products in gas and liquid phase of the EOR reaction system by on-line GC, ion chromatography (IC) and ¹H NMR (Supplementary Fig. 26). It is found out that acetate is only detected product in the EOR reaction over PdZn/NC@ZnO, and no CO₂ is formed. This demonstrates that the EOR proceeds the common reactive-intermediate pathway that ethanol is initially oxidized to acetaldehyde and subsequently to acetic acid or acetate in alkaline solution, which will not produce CO₂ as by-products (*Nano Energy*, **77**, 105116 (2020); *J. Am. Chem. Soc.* **133**, 15172-15183 (2011); *ChemSusChem* **2**, 99 – 112 (2009); *J. Am. Chem. Soc.* **142**, 13833-13838 (2020); *Nat. Commun.* **8**, 14136 (2017)).

Supplementary Figure 26 | Detections on the products in gas and liquid phase of EOR over PdZn/NC@ZnO. **a** On-line GC spectrum of the gas phase in EOR. **b** IC spectrum of the liquid phase in EOR **c** ^1H NMR spectrum of the liquid phase in EOR.

Comment 2. Reporting the activity per mass of Pd is not the best way to compare the activity of exposed catalytic site. Ideally, this should be reported per surface atom.

Response: Thank you very much for this valuable suggestion. As you suggested, we have calculated the activity of PdZn/NC@ZnO based on per exposed Pd atoms on the surface of PdZn NPs. Pd dispersion ($D_{Pd}/\%$) is calculated from the following equation:

$$D_{Pd}/\% = \frac{600M_{Pd}}{\rho N_A A_{Pd} d_{PdZn}}$$

where the M_{Pd} is 106.42 g/mol, ρ is the density of Pd (12.02 g/cm³), N_A is the Avogadro constant as 6.02×10^{23} /mol, A_{Pd} is the cross section area of Pd atom (7.93×10^{-16} cm²), d_{PdZn} means the average size of PdZn nanoparticles which is estimated by the HR-TEM images, $d_{PdZn} = 4.75$ nm (*Appl. Catal. B.* 257, 117943 (2019); *In Handbook of Heterogeneous Catalysis*; Wiley-VCH: Weinheim, Germany, 1997; pp 689–770)).

The results are described in the revised manuscript and presented in Supplementary Fig. 19 in the revised supplementary information as below:

p. 9 “It is noteworthy that the catalytic efficiency for EOR can even reach up to 77.51 A/mg_{Pd} based on the surfaced Pd atoms of PdZn NPs in PdZn/NC@ZnO (Supplementary Fig. 19).”

Supplementary Figure 19 | CV curves of PdZn/NC@ZnO based on surfaced Pd atoms on PdZn NPs in PdZn/NC@ZnO. The data is recorded in N₂-saturated 1.0 M KOH and 1.0 M C₂H₅OH at room temperature at scan rate of 50 mV/s.

Comment 3. In Figure 3, the caption has mixed up c. and d. At which potential is the chronoamperometry performed?

Response: Thank you very much for careful reading. We feel sorry for our carelessness, we have carefully checked our manuscript and revised the graph annotation in the manuscript. The chronoamperometry measurements are conducted at -0.30 V vs Ag/AgCl in the solution of 1.0 M KOH and 1.0 M ethanol, and this experimental detail has been added to the graph annotation of Fig. 3d as below:

Figure 3 | Catalytic activity and durability evaluation. **a** CV curves of PdZn/NC@ZnO, Pd_n/NC@ZnO, Pd₁/NC@ZnO, and commercial Pd/C, recorded in N₂-saturated 1.0 M KOH and 1.0 M C₂H₅OH at room temperature at scan rate of 50 mV/s. **b** Mass and specific activities of PdZn/NC@ZnO and commercial Pd/C. **c** ECSA values of PdZn/NC@ZnO, Pd_n/NC@ZnO, Pd₁/NC@ZnO, and commercial Pd/C. **d** Chronoamperometry curves of PdZn/NC@ZnO and commercial Pd/C, recorded at their corresponding peak potentials at -0.30 V vs Ag/AgCl in aqueous solution containing 1.0 M KOH and 1.0 M C₂H₅OH.

Comment 4. I have several concerns about the computations. All computational details are only provided in SI making the reading frustrating. Dispersion is missing (and could affect the results with a ethanol molecule that is binbing essentially through van der wall interactions with the metallic surface). The choice of the PdZn model could be better related to the charaterization. It seems different in the main and in the supplementary material!

Response: Thanks a lot for your suggestion. In the revised version, we have added the computational details to the Method part of main text for convenient reading. We adopt DFT-D

approach to consider the van der Waals interaction in the models. Hence, we re-calculate all the geometries as well as their energies. The computational detail part is reorganized as following:

p. 17-19 “**DFT calculations.** All calculations in this work were performed by DMol3 code⁶⁷. The generalized gradient approximation with the Perdew–Burke–Ernzerhof functional⁶⁸ was selected to deal with the exchange and correlation function. For precisely treating the long-range van der Waals interactions, we employed the empirical correction in the Grimme scheme^{69,70}. The DFT Semi-core Pseudopotentials (DSPP) and double numerical plus polarization (DNP) basis set were adopted for the Pd/Zn and C/H/O/N atoms, respectively. The convergence tolerance of energy and force are 1.0×10^{-5} Ha, and 2.0×10^{-3} Ha /Å during the fully geometry optimization. The vacuum space along z direction is set to be 15 Å to avoid interaction between the two neighboring images. A Monkhorst–pack mesh of $2 \times 2 \times 1$, $2 \times 1 \times 1$ and $1 \times 1 \times 1$ k-points was used in sampling the integrals over the Brillouin zone. For Pd-Pd sites, Pd-Zn dual sites and individual Pd sites related models, respectively⁷¹. Calculated lattice constants for PdZn are 3.003 Å and 3.232 Å, which are in good agreement with them in Materialsprojects (2.895 Å and 3.342 Å)⁷². Slab models of p (3×3) Pd (111) and p (2×3) PdZn (110) were selected as the substrates for ethanol electro-oxidation reaction. Pd-N₄ (pyridine-N₄) embedded in p (6×6) graphene model was selected as the single atom catalyst substrate. The adsorption energies were calculated according to the formula, $E_{ads} = E(adsorbate/sub) - E(adsorbate) - E(sub)$, where $E(adsorbate/sub)$, $E(adsorbate)$ and $E(sub)$ represent the total energy of substrate with adsorbed species, the adsorbate species and the clean substrate. The change in free energy for all the elementary steps are calculated based on the computational hydrogen electrode method developed by Norskov and his co-workers⁷³. the reaction free energy ΔG is defined as the difference between free energies of the final and initial states and is given by the formula:

$$\Delta G = \Delta E + \Delta ZPE - T\Delta S + \Delta G_U + \Delta G_{pH}$$

where ΔE is the DFT calculated reaction energy of reactant and product molecules adsorbed on substrates; ΔZPE and ΔS are the change in zero-point energies and entropy due to the reaction. The bias effect on the free energy of each initial, intermediate and final state involving an electron in the electrode is taken into accounts by shifting the energy of the state by $\Delta G_U = -neU$, where U is the electrode applied potential. ΔG_{pH} is the correction of the H+ free energy at a pH different from 0: $\Delta G_{pH} = -k_B T \ln[H^+] = pH \times k_B T \ln 10$, where k_B is the Boltzmann constant and T is the temperature. The equilibrium potential at pH = 14 was determined to be 0.402 V versus normal hydrogen electrode (NHE) according to the Nernst equation.”

The considered EOR occurring in an alkaline electrolyte (pH = 14) are shown in the following steps^{44, 74, 75}:

To make the PdZn model more related to the characterization, we have supplemented more

calculation studies on the different surface of PdZn NPs in the revised manuscript and supplementary information as follows:

p. 12 “For the surface of intermetallic PdZn model, we compare the surface energies of (001), (100) as well as (110) surfaces, which correspond to the (002), (200) and (110) surfaces observed on PdZn/NC@ZnO by AC HAADF-STEM images (Fig. 1g). The tendency of surface energy is (110) < (100) < (001) (Table S3). Hence, the most stable (110) surface is selected as the reaction surface (Supplementary Fig. 27).”

Table S4. Surface energy of different crystal planes.

Surface	Surface energy(J/m ²)
PdZn(001)	2.13
PdZn(100)	2.07
PdZn(110)	1.88

Supplementary Figure 27 | Geometry of intermetallic PdZn primitive cell as well as the selected (100), (001) and (110) surfaces.

Comment 5. For isolated Pd, the authors should check if OH and EtOH can adsorb on the same site. Currently, the structures that are presented are not convincing. Besides, a Eley-Rideal mechanism cannot be so easily discarded. This is clearly a weakness of the study.

Response: Thank you for these helpful and inspiring suggestions. For individual Pd sites, only one active center is available. Hence there is an adsorption competition between OH and EtOH. Moreover, we determine that the co-adsorption and EtOH and OH can occur due to the steric effect. We calculated the binding energies of all the intermediates considered. The binding energies and adsorption configurations are shown in Supplementary Fig. 28. We can find that intermediates with unsaturated carbon atoms, such as CH₃CHOH, CH₃CO and CH₃COH bind stronger on the individual Pd sites than the intermediates with saturate carbon atoms (CH₃CH₂O, CH₃CHO, CH₃COOH and CH₃CH₂OH). It is noteworthy that the binding strength between CH₃CH₂OH and individual Pd site is quite weak, even weaker than it for water molecule. Furthermore, binding energy for OH is very strong. Hence the ER mechanism should be triggered by the adsorbed OH and the bombarding EtOH molecule. Therefore, we have revised the manuscript to demonstrate the catalytic behavior of individual Pd sites for EOR as below:

Supplementary Figure 28 | DFT calculations on the EOR over individual Pd sites. a Binding energy of adsorbates on modulated individual Pd sites. **b** Adsorption configurations of adsorbates on modulated individual Pd sites.

More exactly, in the first elementary step, the most possible initial configuration should be

constructed by a hydrogen bond from OH* and OH group in EtOH (Supplementary Fig. 29a), instead of from OH* and CH group in EtOH, due to the electronegativity difference between carbon and oxygen elements. After reaction, H₂O* and CH₃CH₂O are formed. Then, the next round of adsorption competition among H₂O, OH and CH₃CH₂O would continue, or the formed CH₃CH₂O would diffuse to the neighboring individual Pd-OH* sites to proceed the next dehydrogenation step based on ER mechanism (Supplementary Fig. 29b). However, the EOR process would be interrupted when CH₃CH₂O is formed. As shown in Fig. R1b, a hydrogen-bond is formed between OH* and oxygen atom in CH₃CH₂O. This configuration cannot trigger the next dehydrogenation step.

Supplementary Figure 29 | DFT calculations on the EOR over individual Pd sites. a The co-adsorption configurations of OH...CH₃CH₂OH and **b** OH...CH₃CH₂O on individual Pd sites.

Therefore, we have revised the manuscript to demonstrate the catalytic behavior of individual Pd sites for EOR as below:

p. 12 “Our theoretical studies suggested that for individual Pd sites the EOR process would be interrupted due to the exclusive hydrogen bond when CH₃CH₂O is formed. (Supplementary Figs. 28 and 29).”

Inspired by this interesting comment, we also considered the possibility of ER mechanism on Pd (111) surface. In the first several steps, bombarding OH would bind on Pd-Pd sites after geometry optimization to make the entire system more stable. Only for CH₃CHO+OH⁻ → CH₂CO + H₂O step, OH could stay above CH₂CHO molecule. The reaction paths of this step along ER and LH mechanisms were shown in Figure R1. A barrier of 17.3 kcal/mol is needed to overcome along LH mechanism, while a higher barrier (44.6 kcal/mol) is needed to overcome along ER mechanism. ER mechanism is not kinetic favorable here.

Figure R1. Reaction profiles for $\text{CH}_3\text{CHO} + \text{OH}^- \rightarrow \text{CH}_2\text{CO} + \text{H}_2\text{O}$ along LH and ER mechanisms on Pd-Pd sites.

It is hard to discard ER mechanism based on experimental method. However, our theoretical studies suggested that for individual Pd sites the EOR process would be interrupted due to the exclusive hydrogen bond when $\text{CH}_3\text{CH}_2\text{O}$ is formed.

Comment 6. The authors claim that they use the CHE method to take into account the potential. But they don't provide the potential at which they computed the reaction profile. And clearly, this is not a potential at which the reaction can proceed since the overall reaction to acetic acid is endothermic in Figure 4. This needs absolutely to be corrected to show results that can indeed be discussed in regard of the experimental results. Besides, the caption of Figure 4 is not coherent with the approach since the authors consider the basic medium in their state but underline the dissociation of a $\text{H}^+ + e^-$ pair in the caption.

Response: This is a great suggestion, the discussion reaction free energy is disordered in the previous version. Considering the your suggestion, we adopt DFT-D to re-calculated all the configurations, total energies, zero-point energies as well as entropies. Hence, in the revised version, we consider the reaction paths based on the updated data from DFT-D.

According to Fig. 3a, we induced ΔG_U under reaction condition ($U = 0.82 \text{ V}$ vs RHE, $\text{pH}=14$) on both Pd and PdZn surface. It was calibrated to U (vs RHE) from $E(\text{Ag}/\text{AgCl})$ by following the formula $U(\text{RHE}) = E(\text{Ag}/\text{AgCl}) + 0.197 + 0.05916\text{pH}$. As shown in Fig. R2, applying $U=0.82 \text{ V}$ making the electrocatalytic elementary step more facile, compared with it under $U = 0.00 \text{ V}$.

Figure 3 | Catalytic activity and durability evaluation. a CV curves of PdZn/NC@ZnO, Pd_n/NC@ZnO, Pd₁/NC@ZnO, and commercial Pd/C, recorded in N₂-saturated 1.0 M KOH and 1.0 M C₂H₅OH at room temperature at scan rate of 50 mV/s.

Figure R2. DFT calculations on the EOR over individual Pd-Pd sites and Pd-Zn dual sites. a,b The free energies profiles of reactions on both Pd-Pd sites ($U = 0.00$ V with respect to the RHE) and c,d Pd-Zn (110) dual sites ($U = 0.82$ V with respect to the RHE) under basic pH = 14. As for the CHE approach, we do not describe it clearly. For sure, in the basic medium, a pH correction should be included. In the framework of CHE, the electronic potential for $H^+ + e^- = 1/2 H_2$ (g) is defined as zero only at standard condition, where the activity of the hydrogen ions must

be 1 (i.e. pH = 0) and the pressure of the hydrogen gas must be 1 bar. If the active of H^+ is not exactly 1, a pH correction should be included. Based on Nernst equation, for the reaction $H^+ + e^- = 1/2 H_2 (g)$, $\Delta G_{pH} = 0.0592pH$ (eV). In our study, we added a ΔG_{pH} (pH = 14) to match the experimental condition. The free energies profiles of reactions on both Pd-Pd sites and Pd-Zn dual sites under acidic (pH = 0) and basic (pH = 14) conditions were shown in Fig. R3. It is clear that increased pH could boost the concentration of OH which make the electrochemical elementary reaction more favorable. The result is in good agreement with the previous theoretical results which suggest that the high alkalinity in this reaction would result in a high coverage of hydroxide group on the metal surface which would facilitate the C-H activation of ethanol to form the adsorbed hydroxyethylidene intermediates (*J. Phys. Chem. C* **113**, 15639–15642 (2009)).

Figure R3. DFT calculations on the EOR over individual Pd-Pd sites and Pd-Zn dual sites. **a,b** The free energies profiles of reactions on both Pd-Pd sites and **c,d** Pd-Zn (110) dual sites under acidic pH = 0 U = 0.00 V and basic pH = 14 U = 0.00 V with respect to the RHE.

In the revised version, we add the free energies profiles of reactions on both Pd-Pd sites and Pd-Zn dual sites under basic (pH = 14) conditions in the main body, as shown in Fig. 4 and Supplementary Fig. 29. We also rewrote the computational detail part in the revised version as below:

p 12 “It can be observed that the reaction procedure of initial dehydrogenation of ethanol (State 2) over Pd-Zn dual sites is easier than that over Pd-Pd sites (-37.88 vs -35.38 kcal/mol). After the 1st dehydrogenation step (State2), CH_3CHOH and CH_3CH_2O are favorable on Pd (111) and PdZn

(110) surface, respectively. The formed CH₃CH₂O could eliminate the possibility of parallel reactions to either CH₃CHO or CH₃COH in the 2nd dehydrogenation step (State3), which coexisted for the 2nd dehydrogenation step of CH₃CHOH on Pd (111) surface. Meanwhile, the desorption processes of acetic acid on the Pd-Zn dual sites are much easier than that on Pd-Pd sites (1.27 vs 7.72 kcal/mol).”

p 17-19 “DFT calculations. All calculations in this work were performed by DMol3 code⁶⁷. The generalized gradient approximation with the Perdew–Burke–Ernzerhof functional⁶⁸ was selected to deal with the exchange and correlation function. For precisely treating the long-range van der Waals interactions, we employed the empirical correction in the Grimme scheme^{69,70}. The DFT Semi-core Pseudopotentials (DSPP) and double numerical plus polarization (DNP) basis set were adopted for the Pd/Zn and C/H/O/N atoms, respectively. The convergence tolerance of energy and force are 1.0×10^{-5} Ha, and 2.0×10^{-3} Ha /Å during the fully geometry optimization. The vacuum space along z direction is set to be 15 Å to avoid interaction between the two neighboring images. A Monkhorst–pack mesh of $2 \times 2 \times 1$, $2 \times 1 \times 1$ and $1 \times 1 \times 1$ k-points was used in sampling the integrals over the Brillouin zone. For Pd-Pd sites, Pd-Zn dual sites and individual Pd sites related models, respectively⁷¹. Calculated lattice constants for PdZn are 3.003 Å and 3.232 Å, which are in good agreement with them in Materialsprojects (2.895 Å and 3.342 Å)⁷². Slab models of *p* (3×3) Pd (111) and *p* (2 × 3) PdZn (110) were selected as the substrates for ethanol electro-oxidation reaction. Pd-N₄ (pyridine-N₄) embedded in *p* (6 × 6) graphene model was selected as the single atom catalyst substrate. The adsorption energies were calculated according to the formula, $E_{ads} = E(adsorbate/sub) - E(adsorbate) - E(sub)$, where $E(adsorbate/sub)$, $E(adsorbate)$ and $E(sub)$ represent the total energy of substrate with adsorbed species, the adsorbate species and the clean substrate. The change in free energy for all the elementary steps are calculated based on the computational hydrogen electrode method developed by Norskov and his co-workers⁷³. the reaction free energy ΔG is defined as the difference between free energies of the final and initial states and is given by the formula:

$$\Delta G = \Delta E + \Delta ZPE - T\Delta S + \Delta G_U + \Delta G_{pH}$$

where ΔE is the DFT calculated reaction energy of reactant and product molecules adsorbed on substeates; ΔZPE and ΔS are the change in zero-point energies and entropy due to the reaction. The bias effect on the free energy of each initial, intermediate and final state involving an electron in the electrode is taken into accounts by shifting the energy of the state by $\Delta G_U = -neU$, where U is the electrode applied potential. ΔG_{pH} is the correction of the H+ free energy at a pH different from 0: $\Delta G_{pH} = -k_B T \ln[H^+] = pH \times k_B T \ln 10$, where k_B is the Boltzmann constant and T is the temperature. The equilibrium potential at pH = 14 was determined to be 0.402 V versus normal hydrogen electrode (NHE) according to the Nernst equation.

Figure 4 | DFT calculated reaction procedure of EOR on Pd-Pd sites and Pd-Zn dual sites. a DFT calculated models of Pd-Pd sites adsorbed with the reactive species from different EOR reaction states. **b** DFT calculated models of Pd-Zn dual sites adsorbed with the reactive species from different EOR reaction states. **c** DFT calculated free energy profiles of EOR over Pd-Pd sites (pH = 14. U = 0.82 V with respect to the RHE). **d** DFT calculated free energy profiles of EOR over Pd-Zn dual sites (pH = 14. U = 0.82 V with respect to the RHE).

Supplementary Figure 29 | DFT-calculated reaction mechanisms of EOR on Pd-Pd sites. For the reactions on Pd-Pd sites associated structure model adsorbed with the reactive species from different states. DFT-calculated EOR free-energies on Pd-Pd sites at pH = 14, U = 0.82 V with respect to the RHE. The purple solid line arrow represents the easy reaction path, and the red dotted line arrow represents the more difficult reaction path.

EOR on Pd-Pd sites undergoes an indirect pathway, i.e., $\text{CH}_3\text{CH}_2\text{OH} \rightarrow \text{CH}_3\text{CHOH}^* \rightarrow \text{CH}_3\text{CHO}^* \rightarrow \text{CH}_3\text{CO}^*$, followed by CH_3CO interaction with surface hydroxyl (OH^*) to form CH_3COOH , agreeing with previous study⁸.

Comment 7. Last, the barriers provided for the CH and OH scission on Pd are much higher than the ones found in the literature (see for instance Neurock work on ethanol oxidation, *Journal of Catalysis* Volume 299, March 2013, Pages 261-271). Proposing barriers greater than 150kJ/mol is clearly unrealistic. I doubt the quality of those results and thus of the overall conclusion about the Zn effect! In figure 4, the TS energy should be also shown in the profiles.

Response: Thanks for your comments. We double check the corresponding barriers based on the DFT-D method. We agree with you that barrier larger than 150 kJ/mol is hard to overcome. The large CH scission in the previous version is mainly caused by the improper CH...OH direction

and the torsion accompanied.

In Neurock's work, the barriers for CH and OH scission in $\text{CH}_3\text{CH}_2\text{OH}$ on Pd with surface hydroxide are 64 and 29 kcal/mol (15.3 and 6.9 kcal/mol), respectively. In our updated data, the barriers for CH and OH scission in $\text{CH}_3\text{CH}_2\text{OH}$ on Pd were 11.8 and 1.7 kcal/mol, respectively. The difference for OH scission is significant. Hence, we test several different initial states, and find that the barriers for OH scission is very low, for sure. In Figure R4, we show different initial states where OH and $\text{CH}_3\text{CH}_2\text{OH}$ are co-adsorbed on Pd-Pd sites, furthermore their corresponding final states and transitions states are also exhibited. The reaction path is along the direction of hydrogen-bond; hence the proton transfer is a low-barrier process (1.7 – 4.4 kcal/mol).

Figure R4. Configurations of initial states, transition states and final states for $\text{CH}_3\text{CH}_2\text{OH} + \text{OH}^- \rightarrow \text{CH}_3\text{CH}_2\text{O} + \text{H}_2\text{O}$ on Pd-Pd sites with different co-adsorption orientations.

We wonder whether the large barrier is caused by a long distance between OH and $\text{CH}_3\text{CH}_2\text{OH}$, then we compare two configurations of OH and $\text{CH}_3\text{CH}_2\text{OH}$ co-adsorbed on Pd-Pd sites with O-H...O hydrogen bond formed and broken by translating $\text{CH}_3\text{CH}_2\text{OH}$ molecule. We find that the configuration with O-H...O hydrogen bond broken is 12.6 kcal/mol higher than it with O-H...O hydrogen bond formed (Fig. R5). The weak binding energy (-22.8 kcal/mol) between $\text{CH}_3\text{CH}_2\text{OH}$ and Pd-Pd sites surface together with its low diffusion barrier (4.3 kcal/mol as show in Fig. R6) determine that the potential surface of $\text{CH}_3\text{CH}_2\text{OH}/\text{Pd-Pd}$ sites is rather flat. Moreover, the $\text{CH}_3\text{CH}_2\text{OH}$ tends to form the O-H...O hydrogen bond with OH^* on Pd-Pd sites to achieve a more stable co-adsorption configuration.

Figure R5. Configurations of OH and CH₃CH₂OH co-adsorbed on Pd-Pd sites with O-H...O hydrogen bond formed (left) and broken (right).

Figure R6. Configurations of initial states, transition states and final states for CH₃CH₂OH diffusion on Pd-Pd sites.

As mentioned above, we determine OH scission along hydrogen bond direction is facile with very low barrier. Furthermore, updated result of CH scission in CH₃CH₂OH on Pd-Pd sites with dispersion correction are in good agreement with Neurock's work.

In the revised version, we adopt reaction free energy diagram instead of the relative energies diagram to illustrate the performances of different substrates in EOR based on computational hydrogen electrode method, which has been presented in Fig. 4 as below:

Figure 4 | DFT calculated reaction procedure of EOR on Pd-Pd sites and Pd-Zn dual sites. a DFT calculated models of Pd-Pd sites adsorbed with the reactive species from different EOR reaction states. **b** DFT calculated models of Pd-Zn dual sites adsorbed with the reactive species from different EOR reaction states. **c** DFT calculated free energy profiles of EOR over Pd-Pd sites (pH = 14, U = 0.82 V with respect to the RHE). **d** DFT calculated free energy profiles of EOR over Pd-Zn dual sites (pH = 14, U = 0.82 V with respect to the RHE).

Reviewers' Comments:

Reviewer #1:

None

Reviewer #2:

Remarks to the Author:

the authors answered my questions well and revised the writing properly. I have no further comments.

Reviewer #3:

Remarks to the Author:

The authors have addressed my concerns and considerably improved the manuscript that is now reaching the standard of Nature communications.

Reviewer #2:

The authors answered my questions well and revised the writing properly. I have no further comments.

Response: We appreciate your recommendation of acceptance and helpful comments in the reviewing process and are pleased to have our manuscript be reviewed by you.

Reviewer #3:

The authors have addressed my concerns and considerably improved the manuscript that is now reaching the standard of Nature communications.

Response: Thank you very much for your valuable comments in the reviewing process, which have helped us to improve the quality of the whole manuscript. We sincerely appreciate you for recommending our manuscript be accepted by *Nature Communications*.